# Brain-infiltrating CD4 T cells drive inflammatory microglia proliferation during cryptococcal meningitis in mice

Sofia Hain [1,8], Man Shun Fu[1,8], Lucy Wigg[2], Lorna George[1], David Lecky[1], Alexander J. Whitehead[3], Erin Clipston[2], Ko Sato[4,5], Masahiro Ono [6], Marcel Wuthrich[3], Bruce Klein[3], Kazuyoshi Kawakami[4], Julie Rayes [7], David Bending [1] & Rebecca A. Drummond [1,2] ✉

Cryptococcal meningitis is a fungal infection in patients with compromised CD4 T cell function. CD4 T cells provide killing signals to macrophages, principally IFNγ, to limit intracellular fungal replication. However, CD4 T cells may also drive inflammatory tissue damage. Yet, it is not fully understood how fungal-specific CD4 T cells infiltrate the brain and how they influence functional phenotypes of CNS-resident myeloid cells. In the current work, we develop a mouse model to track fungal-specific CD4 T cells and determine their influence on microglia. We found IFNγ+ fungal-specific CD4 T cells have limited TCR signalling and characterise a population of inflammatory microglia that upregulate MHCII and IFNγ-regulated genes during infection. Inflammatory microglia have poor fungicidal capacity and significantly expand during infection, a process that depends on CD4 T cell infiltration. Taken together, these data identify the early inflammatory consequences of fungal-specific CD4 T cell infiltration and identify proliferating microglia as important drivers of brain inflammation during infection.

Cryptococcal meningitis is the leading cause of fungal brain infection in humans with acquired immunodeficiency syndrome (AIDS), causing ~100,000 deaths each year[1]. The causative agent, *Cryptococcus neoformans*, is an environmental fungus that has a striking ability to evade myeloid-based host defences and can replicate intracellularly within macrophage phagosomes[2]. *C. neoformans* secretes several immune-modulating factors to dampen expression of antimicrobial enzymes such as inducible nitric oxide synthase (iNOS). This helps prevent fungal killing as macrophages largely depend on IFNγ-mediated activation and generation of toxic molecules including nitric oxide (NO) to destroy intracellular fungi[2,3]. Yet, *C. neoformans* immune modulation

does not affect macrophage subtypes equally, with some macrophage subsets exhibiting a profound susceptibility to *C. neoformans* intracellular infection, and others more resistant[4,5]. For example, alveolar macrophages do not become heavily infected with the fungus whereas interstitial macrophages, particularly the MHCII[hi] subset, are the major intracellular niche for *C. neoformans* in the lung[4,6].

Deficiencies in CD4 T cells or IFNγ signalling (e.g. mutation in *IL12RB1*) result in an enhanced susceptibility to *C. neoformans* infection[2,7]. This is due to a lack of IFNγ-mediated activation of macrophages, enhancing intracellular residence of the fungus and reducing fungal killing and clearance. Indeed, patients with HIV-associated

[1]Institute of Immunology & Immunotherapy, University of Birmingham, Birmingham, UK. [2]Institute of Microbiology & Infection, University of Birmingham, Birmingham, UK. [3]Department of Pediatrics, University of Wisconsin School of Medicine and Public Health, University of Wisconsin-Madison, Madison, WI, USA. [4]Department of Microbiology, Mycology and Immunology, Tohoku University Graduate School of Medicine, Aoba-ku, Sendai, Japan. [5]Department of Clinical Microbiology and Infection, Tohoku University Graduate School of Medicine, Aoba-ku, Sendai, Japan. [6]Department of Life Sciences, Imperial College London, London, UK. [7]Department of Cardiovascular Sciences, College of Medicine and Health, School of Medical Sciences, University of Birmingham, Birmingham, UK. [8]These authors contributed equally: Sofia Hain, Man Shun Fu. ✉e-mail: r.drummond@bham.ac.uk

cryptococcal meningitis that have been treated with adjunctive IFNγ therapy have reduced fungal burden in the cerebrospinal fluid[8,9], indicating that replacement of IFNγ in patients lacking CD4 T cells may help to control *C. neoformans* infection. Better management of HIV infection with antiretroviral therapy (ART) has also reduced the incidence of *C. neoformans* infections in recent years, by helping patients regain immune function and boosting circulating CD4 T cell counts[1]. However, in ~10–30% patients, this regained immune function is paradoxically accompanied by new or worsening *C. neoformans* infection and brain inflammation. This condition, called immune reconstitution inflammatory syndrome (IRIS), is thought to be caused by excessive CD4 T cell activation to fungal antigens resulting in inflammatory tissue damage in the brain[10].

During *C. neoformans* infection, the influence of infiltrating CD4 T cells on brain-resident myeloid cells and how they actively contribute towards the resulting inflammation is unclear. This is partly because the functional roles and phenotypic diversity of tissue-resident microglia, inflammatory macrophages and recruited monocytes during *C. neoformans* brain infection is poorly defined. Yet, this is important to understand in the context of recent findings that described distinct activation states of brain-resident microglia and macrophage populations that arise in response to inflammation[11,12]. The brain is populated with several functional subsets of myeloid cells, including tissue-resident microglia, perivascular macrophages, and other border-associated macrophages[13]. During infection or sterile inflammation, these subsets can further diversify into additional subsets and phenotypes[11,12], and may also be accompanied by monocyte-derived inflammatory macrophages that are recruited from the blood or neighbouring sites such as the meninges[14]. Understanding how this myeloid network is shaped and modulated during different disease states has enabled the identification of novel mediators of tissue damage in the brain, revealing potentially useful therapeutic targets. For example, microglia differentiate into a lipid-metabolising inflammatory activation state termed damage-associated microglia (DAMs) which has been identified in Alzheimer's and other CNS disorders[11,15].

In the current work, we developed tools to study antigen-specific CD4 T cell recruitment and signalling during acute *C. neoformans* brain infection. We used an acute infection model with a virulent strain of *C. neoformans* to focus on the early phases of CD4 T cell activation in this tissue. We then characterise the phenotypic diversity of brain myeloid cells during *C. neoformans* infection, discovering a small population of inflammatory microglia that expands with infection and expresses genes indicative of CD4 T cell interaction. These inflammatory microglia expanded in a manner that was dependent on CD4 T cell brain infiltration and contributes towards brain inflammation during infection.

## Results

### CD4 T cells accumulate in the brain during acute *C. neoformans* infection

To evaluate the dynamics of CD4 T cell infiltration to the brain during *C. neoformans* infection, we used an acute infection model with the serotype A *C. neoformans* strain H99, the best studied reference strain used in animal models of cryptococcal meningitis[16]. When injected intravenously, *C. neoformans* H99 rapidly colonises the brain and establishes meningitis and brain infection, with little variability in fungal brain burden between animals (Fig. 1a). Infected immune-competent C57BL/6 J mice exhibit little weight loss in this infection until after day 6 post-infection and begin to succumb to the infection around day 9 post-infection, eventually resulting in 100% mortality (Fig. 1b). We therefore defined days 1-4 post-infection as 'early', and days 6-9 post-infection as 'late'. In this model, we found CD4 T cells accumulated in the infected brain from day 6 post-infection, increasing in number as infection progressed (Fig. 1c). Indeed, mice infected with increasing doses of *C. neoformans* demonstrated a strong positive correlation between number of brain-infiltrating CD4 T cells and brain

fungal burden (Fig. 1d). We confirmed that CD4 T cells in the brain were within the tissue and not contamination from the peripheral blood, as mice injected intravenously with the cell labelling dye CFSE prior to analysis had <10% labelled CD4 T cells in the brain following perfusion (Fig. 1e). Microscopy analysis identified CD4 T cells accumulating near sites of extracellular yeast growth in the brain (Fig. 1f) and were found in ~77% of infection sites analysed (*n* = 26 total sites analysed; 20 with CD4 T cells present). Infection sites were observed in the brain cortex, thalamus, nasal bulb and cerebellum regions (Fig. S1). To determine if infiltrating CD4 T cells were fungal-specific, we stained lymphocytes from uninfected and infected brains using an MHCII tetramer loaded with peptide derived from *C. neoformans* protein Cda2, an immuno-dominant antigen for this fungus[17]. This experiment revealed that antigen-specific CD4 T cells infiltrated the *C. neoformans*-infected brain and were not found in uninfected brains (Fig. 1g). Taken together, this data shows that CD4 T cells, including antigen-specific populations, infiltrate the *C. neoformans* infected brain during the late stages of acute infection and positively correlate with fungal brain burden.

### Fungal-specific CD4 T cell brain infiltration coincides with establishment of high fungal burdens

We next interrogated the dynamics of the antigen-specific CD4 T cell response in the brain during *C. neoformans* infection. MHCII tetramers loaded with fungal peptides detected the presence of antigen-specific CD4 T cells in the brain (Fig. 1g), but small numbers and the rarity of these cells precluded any extensive analysis of this population. Therefore, we used an adoptive transfer model (Fig. 2a), which increases the number of fungal-specific CD4 T cells in circulation and enables functional analysis of these lymphocytes within the timeframe of an acute infection. We performed adoptive transfers of fungal-specific TCR transgenic CD4 T cells that specifically recognise a peptide from Cda2[18], an immunodominant antigen for *C. neoformans*[17]. We then used flow cytometry to track these TCR transgenic T cells (termed CnT.II, see Fig. S2 for gating strategy) to the brain during acute *C. neoformans* infection and analysed functional parameters including proliferation, cytokine production and expression of activation markers (Fig. 2b–e). At early stages of infection (day 4), we found a low frequency of CnT.II cells in the brain and were unable to accurately determine proliferation rates or expression of activation markers at this time point, although we did detect a small fraction of proliferating CnT.II cells in the lung, spleen and cervical lymph nodes at this time point (Fig. S3). The lack of CnT.II proliferation and recruitment to the brain was likely related to the lower fungal burdens at this day 4 post-infection, since we found that infiltration of CnT.II cells to the brain was highly influenced by fungal brain burden (Fig. 2c), similar to total CD4 T cells (Fig. 1d), which indicated that the higher fungal burdens observed at later stages of infection (i.e. after day 6, Fig. 1a) were required prior to the infiltration of fungal-specific CD4 T cells to the brain. Indeed, we found significant increases in the frequency, proliferated and activation of CnT.II cells in the brain, lung, spleen and brain-draining cervical lymph nodes at day 7 post-infection, compared to uninfected controls (Fig. 2b, d, e), as well as increased infiltration of CnT.II cells in the meninges (Fig. 2f), which are a major gateway to the brain for infiltrating lymphocytes[19]. Therefore, these data show that fungal-specific CD4 T cell infiltration to the brain occurs after high fungal burdens have been established.

### Infiltrating fungal-specific CD4 T cells are IFNγ-producing effector cells

Since fungal-specific CnT.II cells did not infiltrate the brain until later stages of infection, we focussed our studies of the functional phenotype of these cells to day 7 post-infection. We found that the majority of CnT.II T cells recruited to the infected brain had undergone several rounds of division, since these cells were negative for proliferation dye compared to CnT.II cells in the draining cervical lymph nodes, spleen

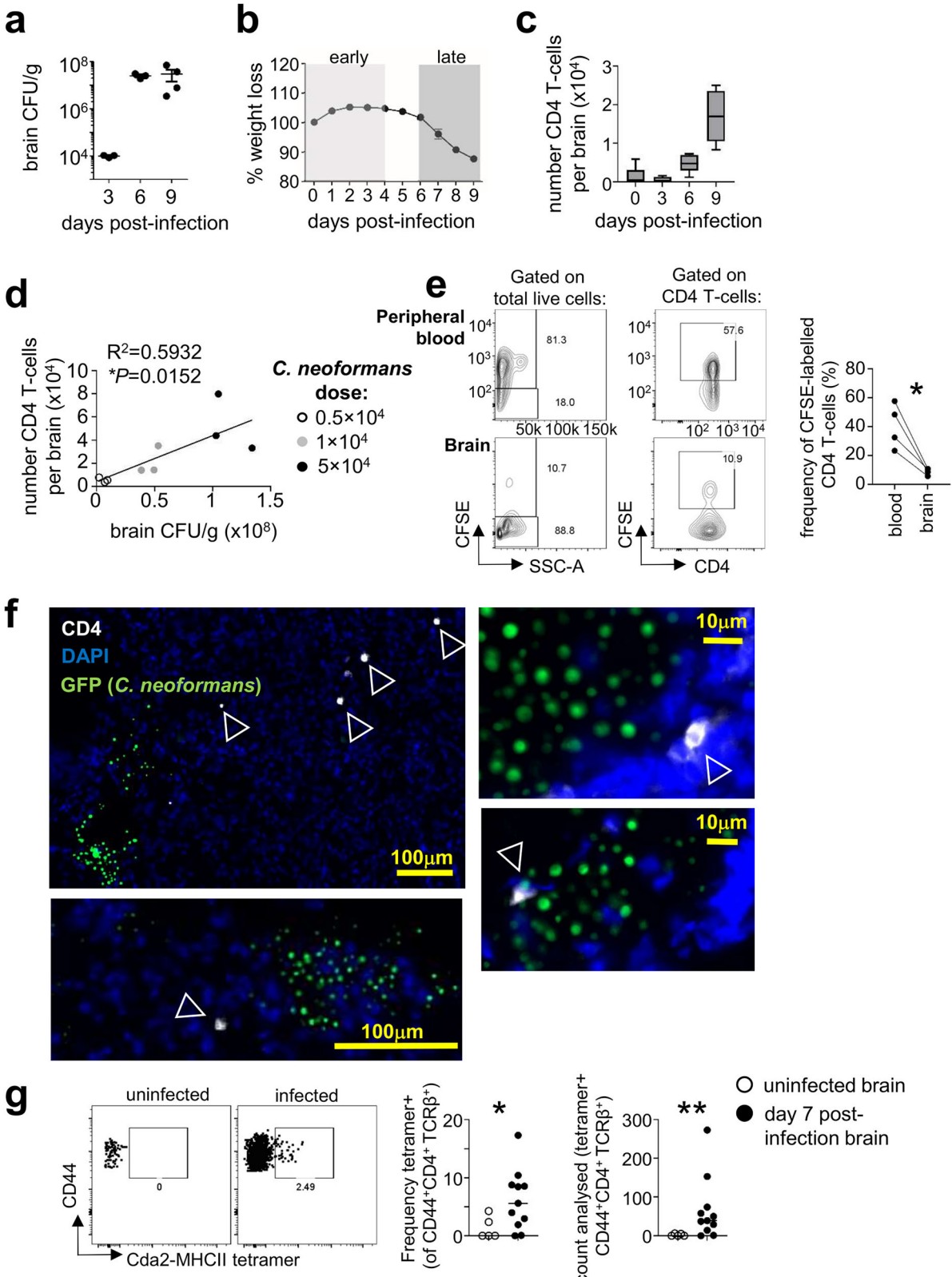

and lung, which had moderate levels of proliferation by day 7 post-infection (Fig. 3a). Brain-infiltrating CnT.II cells produced IFNγ and highly expressed activation marker CD44, with the majority of cells exhibiting an effector functional phenotype (Fig. 3b, c). We did not find any evidence for IL-17A production by brain-infiltrating CnT.II cells (Fig. 3d). Taken together, our data show that fungal-specific CD4 T cells are recruited to the *C. neoformans* infected brain late post-

infection, after undergoing several rounds of cell division and differentiating into IFNγ-producing effector cells.

### Fungal-specific CD4 T cells have limited TCR signalling in the brain

We next examined TCR signalling by *C. neoformans*-specific CD4 T cells recruited to the infected brain, to gain initial insights into the dynamics

**Fig. 1 | CD4 T cells infiltration to the *C. neoformans* infected brain correlates with fungal burden and accumulate near sites of yeast growth. a** Fungal burdens in the brain at days 0, 3, 6 and 9 post-infection. Data from a single experiment, *n* = 3 mice (day 3 and 6) or 4 mice (day 9). Data presented as mean +/- SEM. **b** Weight loss, relative to starting weight, following i.v. infection with *C. neoformans* H99 in C57BL/6 mice. Data pooled from two independent experiments, *n* = 11 mice per time point. Data show as mean +/- SEM. **c** Total number of CD4 T cells per brain at days 0 (*n* = 5 mice), 3 (*n* = 4 mice), 6 (*n* = 7 mice) and 9 (*n* = 6 mice) post-infection. Data pooled from 1 to 2 independent experiments. Whiskers refer to the maximum/minimum values, the box refers to interquartile ranges, the centre line refers to the mean. **d** Number of brain-infiltrating CD4 T cells relative to brain fungal burden in mice infected with low (open symbol), medium (grey symbol) and high (filled symbol) dose of *C. neoformans*. Each point represents an individual animal. Data analysed by simple linear regression. **e** Example flow cytometry plots of peripheral

blood and brain samples (matched from same animal) from mice pre-injected with cell labelling dye CFSE 10 min prior to euthanasia and analysis. Graph shows quantification where each point represents an individual animal. Data analysed by paired t-test. *\*P* = 0.0206. **f** Example confocal microscopy images of CD4 staining in the infected brain at day 7 post-infection (representative of 3 animals analysed). Wild-type mice were infected with GFP-expressing *C. neoformans* and brain sections stained with DAPI (blue) and anti-CD4 (white). **g** Example Cda2-MHCII tetramer staining of leukocytes isolated from uninfected (open symbol) or *C. neoformans*-infected brains (at day 7 post-infection; closed symbol). Example dot plots are pre-gated on CD45$^{hi}$CD44$^+$CD4$^+$TCRβ$^+$ live singlets. Frequency and the number of cells acquired were quantified in uninfected (*n* = 5 mice) and infected (*n* = 11 mice) animals. Data pooled from two independent experiments and analysed by Mann-Whitney U-tests. *\*P* = 0.0389, *\*\*P* < 0.0094.

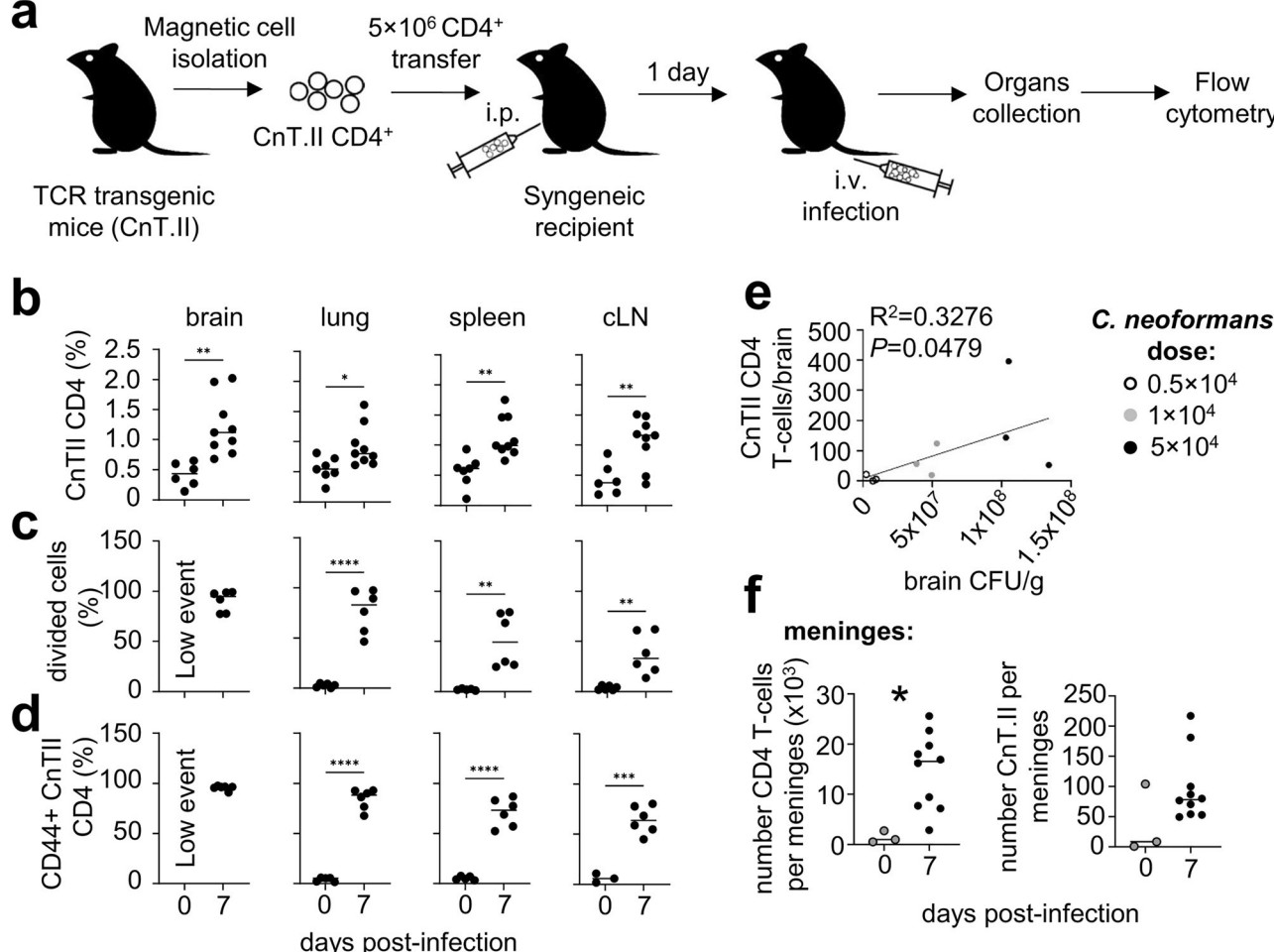

**Fig. 2 | Fungal-specific CD4 T cells are recruited to the brain late post-infection. a** Schematic of adoptive transfer model used in this study. **b** Frequency of transferred fungal-specific CnT.II CD4 T cells in the brain (*n* = 6 mice day 0, *n* = 9 mice day 7, *\*\*P* = 0.0021), spleen (*n* = 7 mice day 0, *n* = 9 mice day 7, *\*\*P* = 0.0021), lung (*n* = 7 mice day 0, *n* = 9 mice day 7, *\*P* = 0.0212) and cLN (*n* = 6 mice day 0, *n* = 9 mice day 7, *\*\*P* = 0.0052), **c** the frequency of divided (CFSE$^{low}$) CnT.II cells in the brain (*n* = 6 mice day 7), spleen (*n* = 5 mice day 0, *n* = 6 mice day 7, *\*\*P* = 0.0027), lung (*n* = 5 mice day 0, *n* = 6 mice day 7, *\*\*\*\*P* < 0.0001) and cLN (*n* = 6 mice day 0, *n* = 6 mice day 7, *\*\*P* = 0.0024) and (**d**) frequency of CD44 expression within CnT.II cells in the brain (*n* = 6 mice day 7), spleen (*n* = 5 mice day 0, *n* = 6 mice day 7, *\*\*\*\*P* = 0.0001), lung (*n* = 5 mice day 0, *n* = 6 mice day 7, *\*\*\*\*P* = 0.0001) and cLN

(*n* = 3 mice day 0, *n* = 6 mice day 7, *\*\*\*P* = 0.0002). Data is pooled from two independent experiments and analysed by two-way ANOVA. **e** Number of brain-infiltrating CnT.II CD4 T cells relative to brain fungal burden in mice infected with low (open symbol), medium (grey symbol) and high (closed symbol) dose of *C. neoformans*. Each point represents an individual animal. Data analysed by simple linear regression. **f** Total number of CD4 T cells and fungal-specific CnT.II T cells in the meninges of uninfected (*n* = 3 mice; open symbol) and infected (**n** = 10 mice; day 7 post-infection, closed symbol) mice, Data pooled from three independent experiments (one uninfected mouse per experiment) and analysed by unpaired t-test. *\*P* = 0.0127.

of T cell interactions with MHCII-expressing myeloid cells within this tissue. For that, we crossed CnT.II TCR transgenic mice with *Nr4a3*-Timer-of-cell-kinetics-and–activity (Tocky) reporter mice[20]. In these mice, *Nr4a3* (a distal TCR signalling protein) regulatory elements drive

expression of a fluorescent timer protein that initially emits a blue fluorescence and then spontaneously switches to a red fluorescent form after the maturation half-time of 4 h[20,21]. These *Nr4a3*-Tocky mice can therefore be used to determine the timing of TCR signalling by

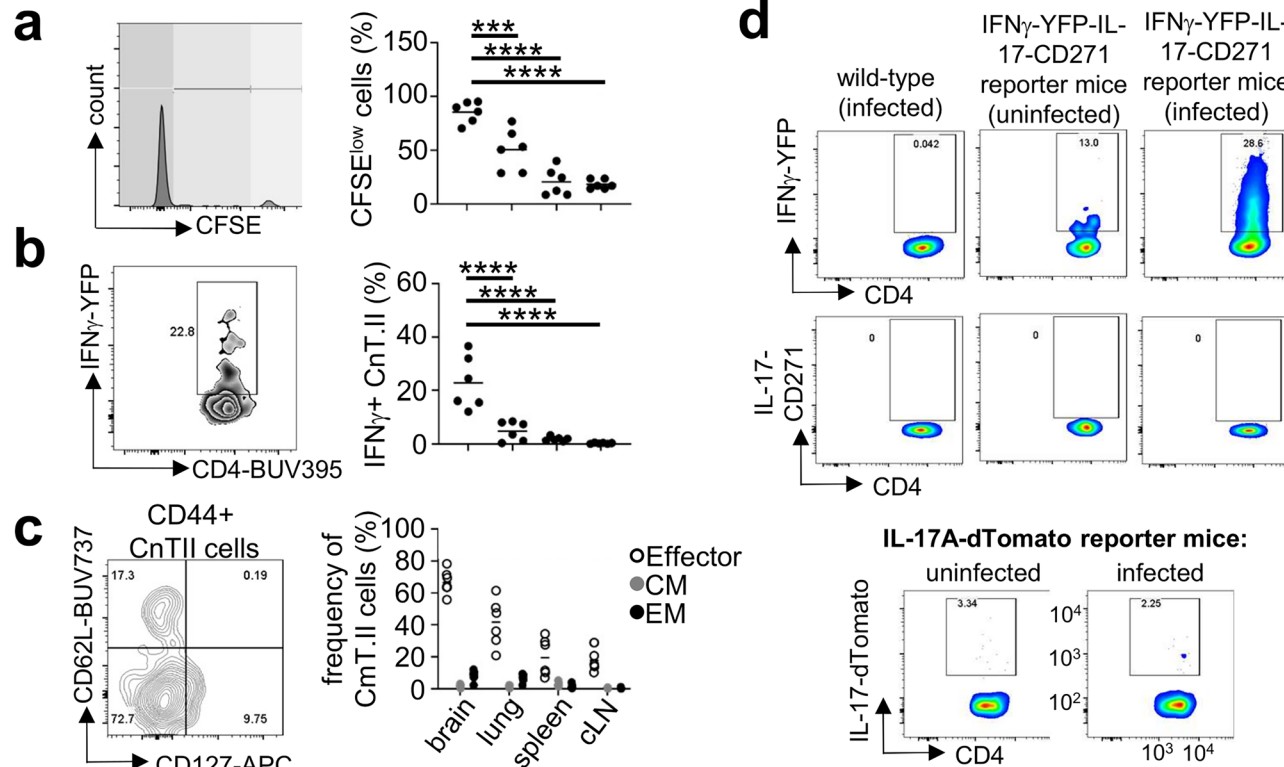

**Fig. 3 | Brain-infiltrating fungal-specific CD4 T cells are IFNγ-producing effector cells. a** Frequency of CnT.II cells that had diluted out cell proliferation dye indicating a high number of cell divisions shown by dark grey gate shown at far left of example flow plot, $n = 6$ mice analysed per organ, **b** the production of IFNγ by CnT.II cells ($n = 6$ mice analysed per organ) and **c** the proportion of CnT.II cells with an effector (CD44+CD127−CD62L−; open symbol), central memory (CM; CD44+CD127+CD62L+; grey symbol) or effector memory (EM; CD44+CD127−CD62L−; closed symbol) phenotype in the indicated organs at day 7 post-infections ($n = 6$ mice analysed per organ). CLN cervical lymph nodes. Data pooled from 2

independent experiments. Each point represents an individual mouse, with the line representing the mean. Data analysed by two-way ANOVA. ***$P = 0.0007$, ****$P < 0.0001$. **d** Example flow cytometry plots of IFNγ-YFP and IL-17A-CD271 reporter expression in wild-type (no reporter genes), and uninfected and infected (day 7 post-infection) reporter animals. Similar results were also confirmed in a separate IL-17A-dTomato reporter mouse line, and example flow cytometry plots of IL-17A-dTomato reporter expression is shown for uninfected and infected (day 7 post-infection) reporter animals. Results are representative of 9 animals, analysed in three independent experiments.

comparing the ratio of blue (new TCR signalling) and red (historical TCR signalling) fluorescence. CnT.II-*Nr4a3*-Tocky mice were used as donors for our adoptive transfer experiments as before, and the expression of *Nr4a3*-Tocky was compared by fungal-specific CD4 T cells recruited to the brain, lung and lymphoid organs. Despite high fungal burdens within the brain (Fig. 4a), we found expression of *Nr4a3*-Tocky was limited to ~15% of infiltrating CnT.II T cells, compared to ~25-30% of CnT.II recruited to other organs (Fig. 4b). This was not because CnT.II-*Nr4a3*-Tocky cells were inherently defective in TCR signalling, since we could induce expression of *Nr4a3*-Tocky following in vitro stimulation with Cda2 peptide-loaded splenocytes (Fig. 4c), and mice injected with Cda2 peptide upregulated expression of *Nr4a3*-Tocky in both the brain and spleen in vivo, albeit only at high peptide doses (Fig. 4d).

Although Cda2 is an immunodominant antigen for *C. neoformans*, previous studies indicated that antigen-specific responses to *C. neoformans* are highly polyclonal[17]. Since we detected limited expression of *Nr4a3*-Tocky by brain-infiltrating Cda2-specific CnT.II CD4 T cells, we decided to analyse total CD4 T cell responses to determine if limited TCR engagement was a general feature of CD4 T cells in the fungal-infected brain or reflected a lack of diversity in the TCR repertoire in the CnT.II adoptive transfer model. We directly infected *Nr4a3*-Tocky mice which have a normal TCR repertoire and analysed the expression of *Nr4a3*-Tocky in the brain, lung and lymphoid organs. In these experiments, we detected *Nr4a3*-Tocky signal in ~30% of total brain-infiltrating CD4 T cells after day 7 post-infection which remained at this level until day 9 post-infection (Fig. 4e). Similar dynamics were

observed in the spleen and lung (Fig. 4e), although there was less Nr4a3-Tocky in the blue form (i.e. new TCR signalling) in these tissues (Fig. 4e). These data indicated that TCR engagement by brain-infiltrating CD4 T cells requires a high antigen load, since it could only be detected when analysing populations responding to multiple antigens. As our adoptive transfer experiments indicated that entry to the brain was restricted to activated effector CD4 T cells, we measured surface expression of the TCRβ chain since downregulation of the TCR is a common feature of activated effector cells. Compared to the lung and spleen, CD4 T cells recruited to the brain had significantly reduced surface expression of TCRβ (Fig. 4f). Therefore, CD4 T cells recruited to the brain only engage their TCR when exposed to high antigen levels, in part due to lower expression of the TCRβ as a result of their activation status.

## An inflammatory microglia subset expressing MHCII and IFNγ-regulated genes expands during *C. neoformans* infection

The activation status of myeloid cells exposed to *C. neoformans* determines infection outcome. Protection against *C. neoformans* infection in humans is mediated by IFNγ-dependent activation of macrophages, where IFNγ derives from CD4 T cells infiltrating into infected tissues[2]. We confirmed that the main cellular source for IFNγ in the *C. neoformans* infected brain was CD4 T cells (Fig. S4). In the brain, there are multiple subsets of CNS-resident myeloid cells including parenchymal microglia and border-associated macrophages[13]. Many of these subsets express the same surface markers, making their distinction by flow cytometry and microscopy

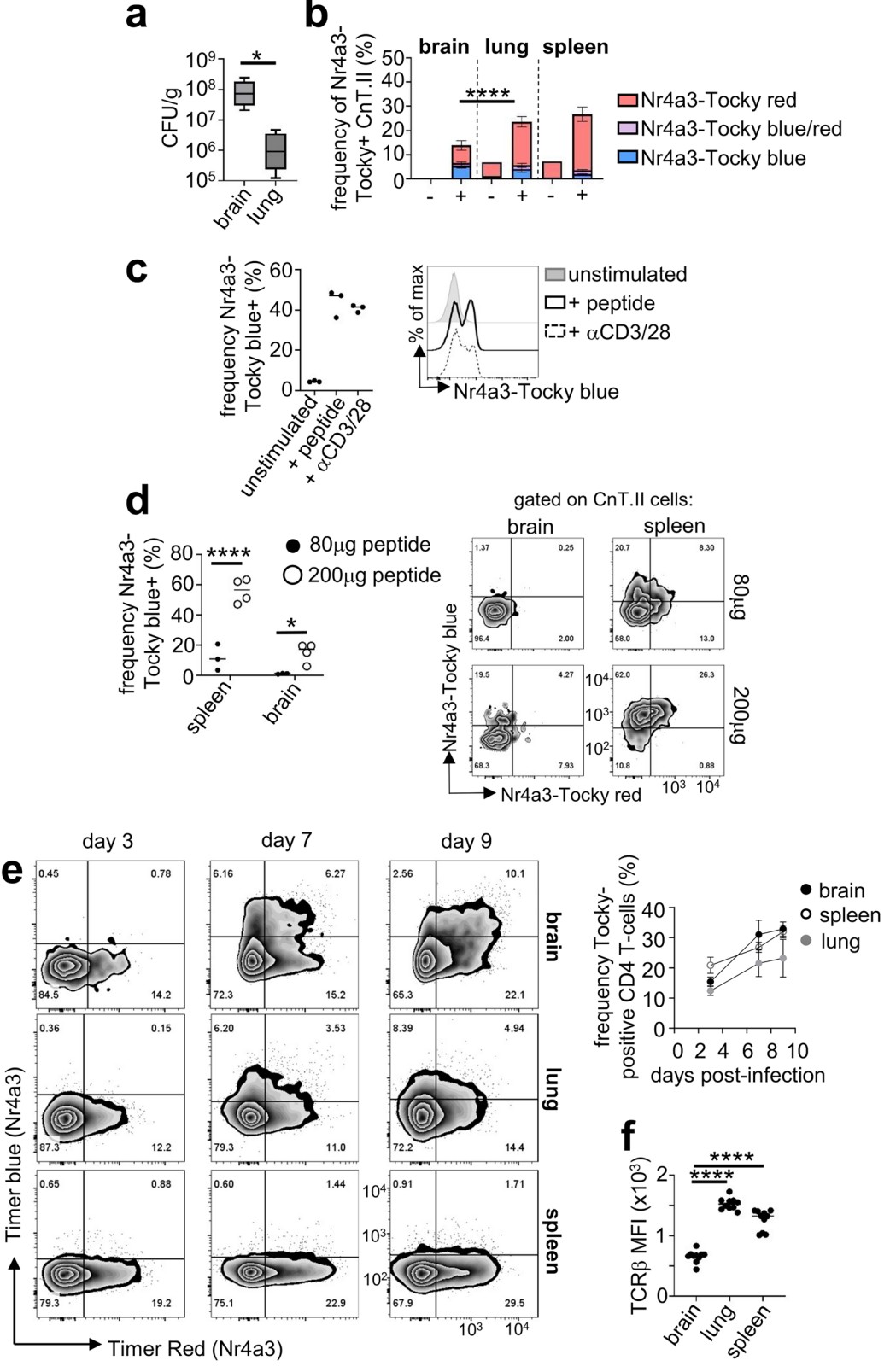

challenging[22]. We therefore used single-cell RNA sequencing to broadly analyse myeloid cell subsets in the *C. neoformans* infected brain. For this, we analysed CD11b[+] myeloid cells from mice at an early (day 3) or late (day 6) stage of infection, compared to uninfected controls. We chose not to analyse mice at the very late stages of infection (e.g. day 9) because flow cytometry indicated that the proportion of microglia was much lower at this time point, which would

have compromised our ability to distinguish rare subsets of these cells (Fig. S5a). Across the three time points (in which 2-5 mice were analysed per time point), we sequenced ~15,000 cells. Graph-based clustering and visualisation using tSNE on this dataset delineated 14 clusters (Fig. 5a). By analysing signature gene expression, we identified 9 of these clusters as microglia and the remainder as monocytes/macrophages and neutrophils (Figs. 5b, S5b).

**Fig. 4 | Fungal-specific CD4 T cells have limited TCR engagement in the brain.**
**a** Fungal burdens in brain and lung at day 7 post-infection of mice (*n* = 6 mice) intravenously infected with *C. neoformans* and adoptively transferred with CnT.II cells as outlined in Fig. 2a. Data pooled from two independent experiments and analysed by Mann Whitney U-test (*P = 0.0189). Whiskers refer to the maximum/ minimum values, the box refers to interquartile ranges, the centre line refers to the mean. **b** Frequency of CnT.II cells expressing Nr4a3-Tocky (blue, red or both forms) in the indicated organs at day 7 post-infection (*n* = 6 mice). Data pooled from two independent experiments, presented as mean +/- SEM and analysed by two-way ANOVA (comparing total Nr4a3-Tocky+ cells), ****P < 0.0001. **c** Frequency of Nr4a3-Tocky+ CnT.II cells that were cultured in vitro either with splenocytes alone (unstimulated), splenocytes loaded with Cda2 peptide or cultured onto plates coated with anti-CD3/28 antibody. Each point represents a technical replicate (*n* = 3 wells). Data is a representative experiment from two independent repeats (each with at least 3 technical replicates). **d** Mice were treated as in Fig. 2a, and then injected with indicated amounts of Cda2 peptide intraperitoneally 4 h prior to analysis (*n* = 3 mice for 80 μg [closed symbol], *n* = 4 mice for 200 μg [open symbol], with line representing the mean). Each point represents an individual animal. Data analysed by unpaired t-test. *P = 0.0479, ****P < 0.0001. **e** Nr4a3-Tocky mice were infected intravenously with *C. neoformans* and the proportion of infiltrating CD4 T cells in the brain (closed symbol), lung (grey symbol) and spleen (open symbol) expressing Nr4a3-Tocky (all forms) determined at days 3, 7 and 9 post-infection. Example plots are gated on total CD3+CD4+ cells in the brain. Data is pooled from two independent experiments, *n* = 3 mice per time point, shown as mean +/- SEM. **f** Mean fluorescence intensity (MFI) of TCRβ expression of CD4 T cells in the brain, lung and spleen of mice (*n* = 10 mice analysed, with line representing the mean) at day 7 post-infection. Data pooled from two independent experiments; each point represents an individual animal and analysed by one-way ANOVA. ****P < 0.0001.

Despite detectable fungal invasion in the brain at day 3 post-infection (Fig. 1a), we found little to no change in the transcriptional response of myeloid cells at this time point when compared with uninfected controls (Fig. S6). In contrast, at day 6 post-infection, we found an accumulation of inflammatory myeloid cells, including monocytes, neutrophils, and monocyte-derived arginase-expressing inflammatory macrophages, in addition to tissue-resident microglia and border macrophage populations (Figs. 5c, S7).

We assigned cluster 11 as inflammatory monocytes based on their increased expression of *Ccr2* and *Ly6c2* (Fig. S5). Cluster 14 was a small, rare population of macrophages that expressed some microglia signature genes (*C1qa, Hexb, Cx3cr1*), albeit at lower levels than microglia (Fig. S5). These cells were transcriptionally similar to choroid plexus macrophages that have been previously described[23]. Cluster 7 contained a larger heterogeneous subset of macrophages. We subdivided this cluster based on divergent expression of MHC Class II, arginase and CD38, which marked out 3 clear populations (denoted 7a, 7b and 7c), that we were also able to replicate using flow cytometry (Fig. S8a–c). Clusters 7a (MHCIIlowCD38+Arginase-) and 7b (MHCIIhighCD38lowArginase-) were present in both infected and uninfected brain (Fig. S8d, e) and were transcriptionally similar to previously described MHCIIlo and MHCIIhi border macrophages[23] (Fig. S8g). Cluster 7b expressed monocyte lineage genes, indicating that these were of monocytic origin whereas 7a did not express monocyte genes (Fig. S8f), in agreement with previously published data on the ontogeny of border macrophages[23]. We therefore assigned cluster 7a as tissue-resident MHCIIlo border macrophages and 7b as the monocyte-derived MHCIIhi border macrophages. Cluster 7c derived almost exclusively from brains isolated at day 6 post-infection (Fig. S8d, e), and expressed high levels of arginase (*Arg1*) (Fig. S8b), as well as high numbers of transcripts encoding monocyte lineage genes (Fig. S8f). We therefore believe cluster 7c is a monocyte-derived inflammatory macrophage, either deriving partly from cluster 7b (MHCIIhi border macrophages) responding to infection and/or from recruited Ly6CHi monocytes that differentiate into macrophages upon arrival to the brain.

Although 9 clusters of microglia were identified in our dataset (Fig. 5b), 8 of these clusters had equal representation in uninfected and infected samples or were distinguished by the expression of immediate-early response genes (IEGs), which are activated by the microglia isolation procedure[24] (Fig. S9a, b), and/or did not separate when visualised with UMAP (Fig. S9c). We therefore believe that these clusters are a result of the limitations in the clustering method used, which can generate biologically irrelevant clusters[25], and are less likely to be biologically relevant in the context of cryptococcosis. We therefore grouped them together in downstream analyses. In contrast, IEGs were not detected in microglia cluster 10 (Fig. S9b), which was predominantly composed of cells from the day 6 post-infection sample (Fig. 5d) and separated away from other microglia clusters using both tSNE and UMAP (Figs. 5a, S9c). This microglia cluster had greater expression of genes encoding pro-inflammatory cytokines and chemokines, pattern recognition receptors, MHC Class II and IFNγ-regulated genes, relative to other microglia in the dataset (Fig. 5e and Supplementary Data 1). Indeed, pathway analysis confirmed that microglia cluster 10 upregulated genes involved with inflammation, antigen presentation and antimicrobial immunity (Fig. 5f). Collectively, this work shows that there is little change to myeloid cell heterogeneity in the early stages of infection, with accumulation of inflammatory myeloid subpopulations, including an inflammatory cluster of tissue-resident microglia, at day 6 post-infection.

## Inflammatory microglia do not resemble damage-associated microglia

We previously found that CNS-resident microglia do not provide protection against *C. neoformans* infection, and instead could harbour the fungus at the early stages of infection[5]. Yet, our sequencing analysis identified a population of inflammatory microglia (cluster 10) that had increased expression of multiple genes (when compared to other microglia in the dataset) that would be predicted to aid in fungal killing, such as IFNγ-regulated guanylate-binding proteins (GBPs) and p47 GTPases which mediate killing of intracellular pathogens[22,26]. We therefore decided to focus our analysis to this microglia subset to better understand its functional role during infection and how CD4 T cells might influence this. First, we compared the pattern of gene expression in the inflammatory microglia subset (cluster 10) to other microglia subsets that have been previously identified in other inflammatory brain conditions. The best characterised microglia subset to date is damage-associated microglia (DAMs), which develop during neurodegenerative diseases and are characterised by Trem2 expression[11]. We compared the pattern of upregulated and downregulated genes in DAMs (relative to their own control) with the pattern of upregulated and downregulated genes in microglia cluster 10 (relative to their own control, i.e. other microglia clusters in the dataset) (Fig. S10a). Although some differentially-expressed genes were shared between DAMs and microglia cluster 10, we found a poor correlation between the two datasets (Fig. S10a).

Since microglia cluster 10 did not appear to resemble DAMs, we next compared to a different microglia subset that has been reported in mice systemically challenged with LPS[12]. In that study, mice were treated with LPS and microglia analysed by single-cell RNA sequencing. Analysis of that dataset identified a cluster of inflammatory microglia, which the authors termed 'inflammation-associated microglia' (IAMs)[12]. In that study, IAMs upregulated genes encoding pattern-recognition receptors and pathways involved in antimicrobial immunity[12]. We performed a similar analysis as before, comparing the pattern of upregulated and downregulated genes in IAMs to microglia cluster 10 in our dataset (Fig. S10b). Here, we found a much stronger correlation between the two microglia clusters, with microglia cluster 10 sharing many of the same upregulated and downregulated genes as IAMs (Fig. S10b). Taken together, these analyses identify microglia

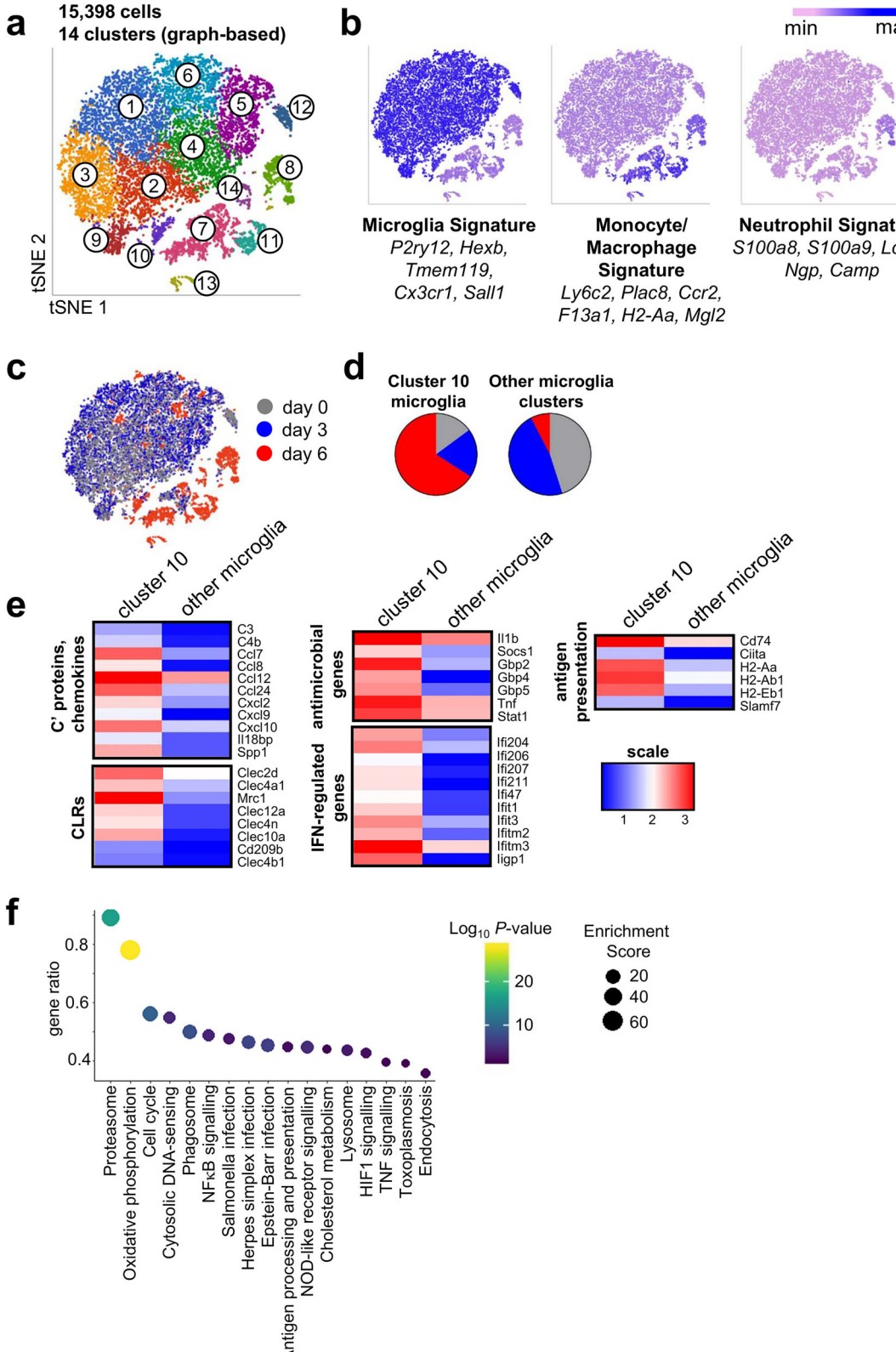

**Fig. 5 | Inflammatory microglia expand during late *C. neoformans* infection.** Analysis of single-cell RNA sequencing dataset from uninfected (*n* = 2) and infected (day 3, *n* = 2; day 6, *n* = 5) animals. **a** tSNE plot showing graph-based clustering results and 14 unique clusters identified within the data. **b** tSNE plots coloured by microglia, monocyte/macrophage or neutrophil signature genes. **c** tSNE plot coloured by time point analysed (day 0, grey; day 3, blue; day 6, red). **d** Proportion of sequenced cells from each of the analysed time points within microglia cluster 10 and all other identified microglia clusters in the dataset. **e** Heatmap of selected genes, grouped by function/similarity, showing mean expression by microglia cluster 10 or other microglia clusters. A full list of differentially-expressed genes is given in Data S1. **f** KEGG pathway analysis of upregulated genes by microglia cluster 10 (compared to other microglia clusters), showing gene ratio of detected genes within indicated pathways, *P*-value and enrichment scores. Statistics derived from Fisher's exact test, with FDR multiple correction.

cluster 10 as more closely resembling IAMs, an activation state of microglia that is distinct of the better characterised DAMs.

## Inflammatory microglia derive from resident microglia

Although IAMs were previously identified from a sequencing dataset[12], no work has been done to establish their origin or function in the context of infection. Unlike DAMs, there is also no published surface markers to enrich for IAMs by other methods, such as flow cytometry. We therefore set out to develop a gating strategy (Fig. S11) for fungal-infection associated IAMs that would allow us to track these inflammatory microglia in the *C. neoformans* infected brain and determine their functional roles during infection. For that, we used our sequencing results to identify potential markers of IAMs for use in flow cytometry experiments to track and isolate these cells in downstream experiments. Our dataset indicated that, amongst the microglia clusters, IAMs highly expressed the mannose receptor (CD206), the scavenger receptor MSR1, and had increased expression of MHCII relative to other microglia (Fig. 6a). Although these genes are also markers of border macrophages, expression levels in IAMs were lower than that of macrophage clusters in our dataset (clusters 7 and 14), which were transcriptionally similar to previously characterised border macrophage populations[23], including perivascular (cluster 7) and choroid plexus (cluster 14) macrophages (Fig. S5). Using these markers, we identified a CD206⁺MSR1⁺ population within our microglia gate (CD45^int CX3CR1^hi) that had increased MHCII expression, aligning with our sequencing results (Fig. 6b). Importantly, this population was significantly increased in the fungal-infected brain compared to uninfected controls (Fig. 6c).

Since the markers we had used to identify IAMs by flow cytometry are also expressed on inflammatory and/or perivascular macrophages, we sought to determine if the CD206⁺MSR1⁺ population was microglia or whether IAMs represented a subset of macrophages that had taken on a microglia-like phenotype. First, we FACS-sorted CD206⁺MSR1⁺CD45^int and CD206⁻MSR1⁻CD45^int cells and performed qRT-PCR for the signature microglia gene *Tmem119*, comparing with CD45^hi CX3CR1⁺ macrophages. We found that CD206⁺MSR1⁺CD45^int cells had increased expression of *Tmem119* compared to macrophages, although this was less than we observed for other sorted CD45^int microglia (Fig. 6d), which mirrored our scRNAseq transcriptional data (Fig. 6e). Indeed, microglia have been shown to downregulate multiple signature genes including *Tmem119* upon activation[27,28]. Next, we infected Sall1^CreER R26^dTomato mice, in which all Sall1⁺ cells express dTomato following Cre-mediated excision of an upstream STOP codon[5]. Sall1 expression is limited to CD45^int CX3CR1^hi microglia in the brain and is not expressed by other types of brain macrophages (Fig. 6f), hence these animals enable specific labelling of the microglia lineage[29]. We found that CD206⁺MSR1⁺ cells were more common within the dTomato-labelled microglia population than the non-labelled population in the fungal-infected brain (Fig. 6g), confirming that the flow cytometry strategy used to identify IAMs (Fig. S11) mostly captured cells of the microglia lineage.

## Inflammatory microglia have poor fungicidal capacity

Pathway analysis indicated that IAMs may be more phagocytic and have greater fungal killing capacity than other microglia, since they highly expressed genes involved with pathogen recognition (*Msr1, Clec4n, Mrc1, Cd14*), phagosome maturation (*Rab7, Atp6v*) and microbial killing (*Il1b, Gbp2*) (Fig. 5e, f). Indeed, MSR1 was recently shown to act as a non-opsonic phagocytic receptor for *C. neoformans*[30]. Moreover, IAMs expressed many genes regulated by IFNγ stimulation (Fig. 5e), including components of the immune proteasome (*Psmb8, Psmb10, Psme1*), and p47 GTPases (*Iigp1, Ifitm3*), which regulate killing of intracellular pathogens in mice[22]. We therefore explored the uptake and killing capacity of IAMs using our flow cytometry gating strategy to identify and isolate these cells. We infected mice with GFP-expressing

*C. neoformans* and measured uptake of yeast cells by CD206⁺MSR1⁺ IAMs and CD206⁻MSR1⁻ microglia. We found that a significantly higher proportion of IAMs were associated with *C. neoformans* yeast compared to other microglia in the brain (Fig. 7a), although both populations with intracellular yeast had similar fluorescence intensities indicating that each population had similar yeast numbers per cell (Fig. 7b). To confirm this, we imaged FACS-purified infected IAMs and non-IAM microglia, then quantified the number of yeast per cell (Fig. 7c). This analysis showed that most microglia had one yeast per cell (68% of IAMs and 89% of non-IAMs). However, we detected a significant increase in microglia hosting 2 and 3 yeast per cell in the IAMs population compared to non-IAMs (Fig. 7d). These results may reflect a superior phagocytic function by IAMs, but may also indicate a higher intracellular infection rate. To determine the difference between these possibilities, we FACS-purified infected (GFP⁺) CD206⁺MSR1⁺ IAMs and infected CD206⁻MSR1⁻ microglia and plated equal numbers onto fungal growth media to determine the viability of the fungus within each population (Fig. 7e). Although there was a slightly higher number of yeast per cell for ~30% of IAMs, we found that IAMs had a significantly higher carriage rate of live fungi than other microglia (Fig. 7e), indicating that these cells had poor fungicidal capacity. In line with that, we found CD206⁺MSR1⁺ IAMs had increased expression of arginase (Fig. 7f), which is typically associated with intracellular fungal infection and poor fungal killing[4].

In uninfected brain, microglia have a stellate-type appearance (Fig. 7g) but exhibit a more rounded morphology upon activation[31]. We used confocal microscopy to examine microglia morphology and location of microglia interacting with *C. neoformans* (Fig. 7h). We found that rounded, activated microglia bordered areas of extracellular yeast growth (Fig. 7h, panels iii and iv), and that these cells often had bound and/or intracellular *C. neoformans* (Fig. 7h, panels v and vi), similar to our previous findings[5].

## IAMs form in response to *Blastomyces dermatitidis* infection

To examine the wider relevance of IAMs to other types of fungal infections, we measured the frequency of IAMs within the brain of mice infected with another fungal pathogen, *Blastomyces dermatitidis*. This dimorphic fungus is endemic to regions of North America and Africa, and can cause intracellular infections of lung macrophages[32]. Central nervous system (CNS) involvement is a complication in ~10% of patients with *B. dermatitidis* infection[33]. We infected mice intravenously with *B. dermatitidis* to model dissemination and used our flow cytometry gating strategy to identify CD206⁺MSR1⁺ microglia as before. These experiments showed that there was a significant increase in CD206⁺MSR1⁺ IAMs in the brains of *B. dermatitidis*-infected mice, which became ill from infection and showed neurological signs of fungal infection (head tilt, circular movements, paralysis in hind limbs) relative to naïve mice (Fig. S12). Therefore, IAMs may represent an inflammatory microglia activation phenotype that broadly develops during fungal infection.

## Inflammatory microglia are highly proliferative

In addition to genes involved with phagocytosis and antigen presentation, our pathway analysis identified 'cell cycle' as a major upregulated pathway in IAMs. Microglia proliferation, or microgliosis, has been shown to drive pathologic brain inflammation and tissue damage in autoimmune CNS disease[34]. We therefore wondered if IAMs were driving inflammation in the *C. neoformans* infected brain via microgliosis, and if this was being influenced by the infiltration of fungal-specific CD4 T cells. First, we analysed whether the cell cycle genes expressed in IAMs were linked to multiple cell cycle stages (indicating active proliferation) or derived from one stage (indicating a cell cycle block). For this, we used a published dataset[35] that sorted mammalian cell cycle genes into G1, S, G2 and M phases, and determined which phase the cell cycle genes expressed by IAMs derived from

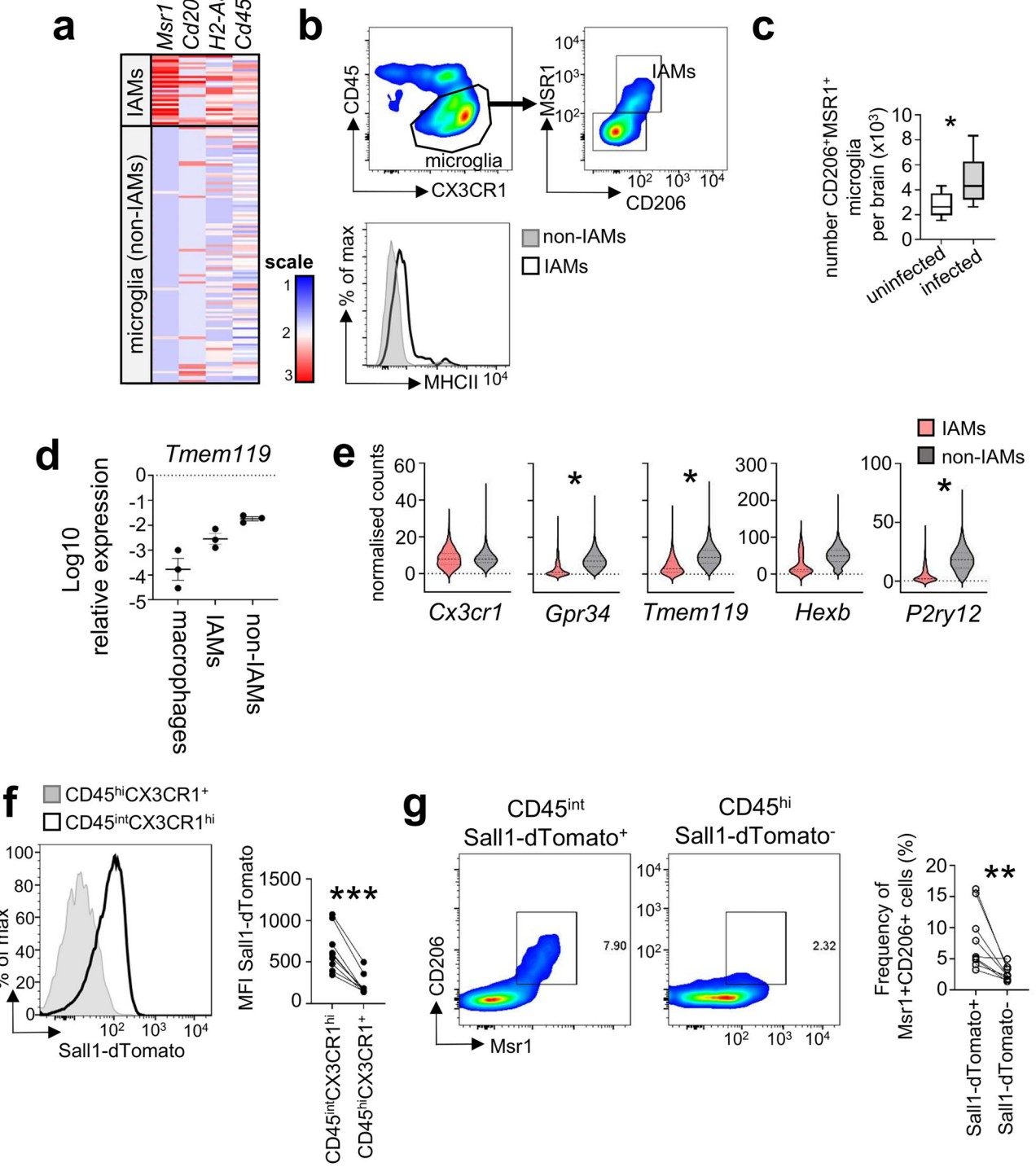

**Fig. 6 | Inflammatory microglia (IAMs) derive from resident microglia and co-express MSR1 and CD206. a** Heatmap of mean expression of indicated genes between IAMs and non-IAMs. **b** Example flow cytometry of IAMs gating. **c** Total number of MSR1+CD206+ microglia in uninfected or infected (day 6 post-infection) brains. Data pooled from 2 independent experiments, $n = 9$ mice per time point. Data analysed by unpaired t-test, *$P = 0.0166$. Whiskers refer to the maximum/minimum values, the box refers to interquartile ranges, the centre line refers to the mean. **d** Quantification of *Tmem119* expression by qRT-PCR on CD206+MSR1+ and CD206-MSR1- microglia (defined as CD45intCX3CR1hi), compared to macrophages (defined as CD45hiCX3CR1+) at day 6 post-infection. Each point represents an independent sort experiment pulled from a batch of mice (10 mice per batch,

$n = 3$ sorts in total), presented as mean +/- SEM. **e** Normalised counts of microglia signature genes in IAMs (red violin) and non-IAMs (grey violin), calculated from the sequencing dataset. Data analysed by Mann Whitney U-tests. *$P < 0.0001$. **f** Example histogram of Sall1-dTomato expression by microglia (CD45intCX3CR1hi) and macrophages (CD45hiCX3CR1+) in *Sall1CreERRosa26Ai14* mice. Graph shows quantification for 8 different animals, from 2 independent experiments. **g** Expression of IAMs markers CD206 and MSR1 within the Sall1-dTomato+ population and Sall1-dTomato- population. The graph shows quantification from 2 independent experiments, where each point represents an individual animal ($n = 8$ mice total). Data in (**f, g**) analysed by paired t-tests. **$P = 0.0055$, ***$P = 0.0004$.

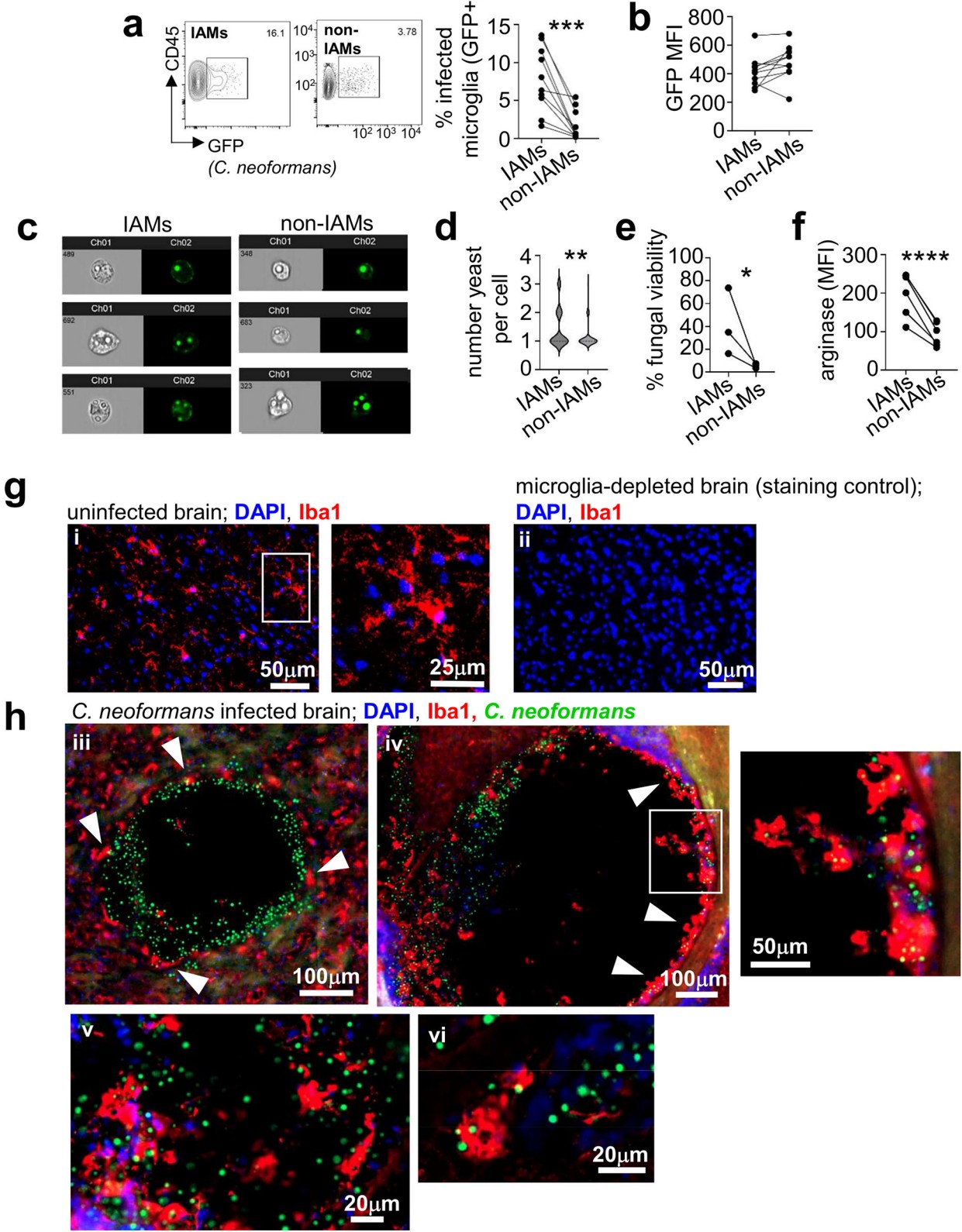

(Supplementary Data 2). This analysis showed that IAMs expressed cell cycle genes across multiple cell cycle phases and were not over-represented by any one phase, indicating that these were actively proliferating cells (Fig. 8a).

Next, we used flow cytometry to determine if IAMs (CD206[+]MSR1[+] microglia) were co-expressing Ki67, a marker of cell proliferation. We found that IAMs were enriched within the Ki67[+] microglia population

(Fig. 8b, d), in line with our sequencing data that indicated IAMs were a proliferating subset. We analysed the frequency of Ki67[+] microglia in the brain at various time-points post-infection using flow cytometry and discovered that the frequency of proliferating microglia significantly increased at day 8/9 post-infection, with the majority of microglia labelling with Ki67 at these time points (Fig. 8c). At day 8 post-infection, these proliferating microglia largely co-expressed IAM

**Fig. 7 | IAMs have poor fungicidal capacity. a** Frequency of IAMs (MSR1$^+$CD206$^+$) and non-IAMs (MSR1$^-$CD206$^-$) associating with GFP-expressing *C. neoformans* at day 6 post-infection as determined by flow cytometry and **b** the MFI of the GFP signal within each population. Each dot represents an individual mouse (*n* = 10 mice analysed), with lines denoting paired samples within the same mouse. Data pooled from 3 independent experiments and analysed by paired t-test, ***$P$ < 0.0007. **c** Example imaging of FACS-purified uninfected microglia, *C. neoformans*-infected IAMs and *C. neoformans*-infected non-IAMs using an imaging flow cytometer (ImageStream). Mice were infected with GFP-expressing *C. neoformans* and microglia isolated at day 7 post-infection for sorting. Sorted populations were fixed and imaged in the brightfield and GFP channels to quantify number of yeast per microglia cell. **d** Quantification of number of yeast per IAM (*n* = 78 microglia counted) and non-IAM (*n* = 53 microglia counted) microglia. Data pooled from two independent sorts/infection experiments and analysed by Mann Whitney U-test. **$P$ = 0.0071. **e** Frequency of live fungal cells within IAMs and non-IAMs. Mice were infected with GFP-expressing *C. neoformans* and infected IAMs or non-IAMs (GFP+) were FACS purified and 500-2000 total cells plated onto YPD agar plates. The number of colonies counted was then expressed as a proportion of the total number of plated cells to calculate viability. Data pooled from 3 independent experiments and analysed by paired t-test, *$P$ = 0.0263. **f** Mean fluorescence intensity of arginase expression (as determined by intracellular flow cytometry) in IAMs and non-IAMs at day 6 post-infection. Each dot represents an individual mouse, lines denote pairing within the same animal. Data pooled from 2 independent experiments and analysed by paired t-test, ****$P$ = 0.0006. **g** Example confocal microscopy of brains from uninfected mice (i, with close up insert) and an uninfected mouse treated with microglia-depleting drug PLX5622[5] as a staining control (panel ii), stained with DAPI (blue) and anti-Iba1 (red). **h** Confocal microscopy of *C. neoformans* infection in the brain at day 7 post-infection. Panels iii and iv show areas of extracellular yeast growth in the cerebral cortex (panel iii) and in the cerebellum (panel iv). White arrows denote rounded, activated microglia. Insert shows close up from panel iv. Panels v and vi show examples of microglia within areas of infection. Example images are representative sections from at least three individual animals in which 3–5 sections per brain were analysed.

markers CD206 and MSR1, however we also observed Ki67 labelling in non-IAM populations in the brain as well (Fig. 8d). Using immunofluorescence, we observed several Ki67$^+$ cells near sites of infection (Fig. 8e) which was increased in the day 8 post-infection brain compared to uninfected (Fig. 8f), aligning with our flow cytometry data. Collectively, these data show that microgliosis occurs during *C. neoformans* infection, and that proliferating microglia mostly map to the inflammation-associated microglia (IAMs) subset.

### Intranasal *C. neoformans* infection induces mild rates of microglia proliferation

*C. neoformans* infections in humans are thought to establish from inhaled spores which germinate into yeast within the lung[36]. Brain infection may occur from this initial inhalation event (thus bypassing the lung), or yeast proliferates in the lungs before escaping into the bloodstream where they travel into the central nervous system, either as free yeast or within infected myeloid cells (Trojan Horse)[36]. Our study had utilised the intravenous route of infection to bypass the lung and initiate acute meningitis rapidly. However, since trafficking of yeast and/or host cells from the lung to the brain following pulmonary infection may alter development of IAMs and microglia proliferation rates, we examined microglia outcomes in mice infected via the intranasal route. Mice infected intranasally have lower fungal brain burdens (~$10^4$ CFU/g, Fig. 8g) than what is achieved with the intravenous route (~$10^7$ CFU/g, see Fig. 1a). Instead, mice succumb to overwhelming lung infection which steadily increases over time (Fig. 8h). Despite the lower brain burdens, we detected a small population of MSR1$^+$CD206$^+$ microglia in the brains of intranasally-infected mice (Fig. 8i), which had higher proliferation rates as detected by Ki67 labelling compared to MSR1$^-$CD206$^-$ microglia (Fig. 8j), similar to what we observed in intravenously-infected mice. However, we observed no increase between day 14 and day 21 in the number of IAMs or total Ki67$^+$ microglia (Fig. 8k), which likely aligned with the little change we saw in brain fungal burden between these time points (Fig. 8g). Taken together, these data show that development of IAMs and microgliosis is significantly influenced by infection route, which impacts on brain infection burden and inflammation within this tissue.

### Infiltrating CD4 T cells are required for microgliosis during *C. neoformans* infection

Previous work has indicated that microglia proliferation and infiltration of autoreactive CD4 T cells to the brain are linked events[34], however, whether proliferating microglia drive T cell infiltration or vice versa is unclear. We therefore examined how loss of lymphocytes affected microglia proliferation and brain inflammation. For that, we infected mice lacking lymphocytes (*Rag2$^{-/-}$*) and measured the number of Ki67$^+$ microglia and IAMs, comparing to wild-type controls.

Strikingly, we found that numbers of proliferating microglia within the *Rag2$^{-/-}$* brain remained similar to uninfected mice whereas microglia in wild-type mice were highly proliferative (Fig. 9a). In line with that, the number of MSR1$^+$CD206$^+$ IAMs was significantly reduced in the *Rag2$^{-/-}$* brain at day 8 post-infection (Fig. 9b). We confirmed this result using an independent strain of *C. neoformans*, Zc1[37]. This strain was isolated from a patient taking part in a clinical trial that ran in Zambia, and is the same molecular type as reference strain H99 (VNI) yet distantly related[37]. Zc1 is slightly less virulent than H99 in mouse infection models as measured by lower fungal brain burdens (Fig. 9c). Wild-type mice infected with *C. neoformans* Zc1 had significantly higher numbers of proliferating microglia than *Rag2$^{-/-}$* mice infected with *C. neoformans* Zc1 (Fig. 9d), as we had observed with reference strain H99 (Fig. 9a). Using immunofluorescence, we observed a significant decrease in Ki67 labelling in brain sections from *Rag2$^{-/-}$* mice compared to wild-type mice at day 8 post-infection (Fig. 9e). Taken together, these data suggested that lymphocytes are important drivers of microglia proliferation during *C. neoformans* infection.

Next, we examined if microglia proliferation during infection was specifically driven by CD4 T cells. For that, we depleted CD4 T cells in wild-type animals prior to infection using a depleting antibody, and then measured microglia proliferation using Ki67 staining as before. These experiments showed that specific depletion of CD4 T cells significantly reduced microglia proliferation during infection compared to isotype-treated controls (Fig. 9f). Importantly, CD4-depleted animals had less weight loss during infection compared to their isotype-treated counterparts (Fig. 9g), which was independent of fungal burden which remained similar between CD4-depleted mice and isotype-controls (Fig. 9h). Taken together, these data show that CD4 T cells are important drivers of microglia proliferation during *C. neoformans* infection, which contributes towards development of clinical symptoms.

## Discussion

Deficiency of CD4 T cells correlates with susceptibility and clinical outcome in patients with cryptococcal meningitis[7]. However, these lymphocytes can also be important drivers of brain inflammation during fungal infection[10]. It is therefore important to understand the dynamics of CD4 T cell responses, particularly fungal-specific populations, and how they influence brain-resident myeloid cells to promote fungal clearance and/or inflammatory responses. Here, we have shown that infiltration of fungal-specific CD4 T cells into the brain during acute infection occurs after the development of significant fungal infection and that there is limited engagement of TCR signalling by these cells in this tissue. Infiltration of these lymphocytes into the brain drove expansion of a proliferating, inflammatory microglia subset (IAMs). These microglia expressed IFNγ-regulated genes

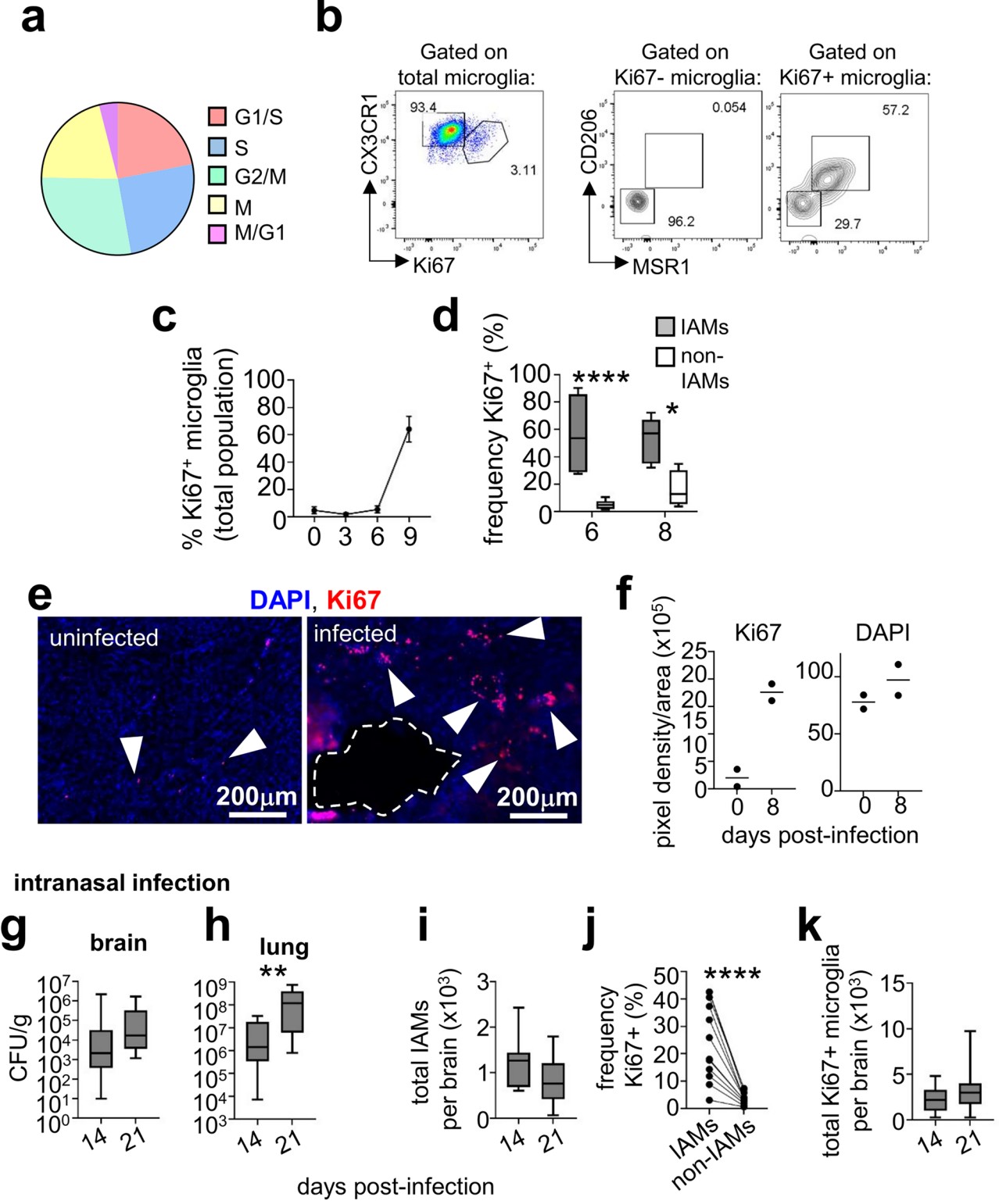

previously linked with antimicrobial immunity, yet remained poorly fungicidal.

Although clinical data demonstrate that CD4 T cells are needed to provide protection against *C. neoformans* infection, it isn't clear which brain-resident myeloid populations most efficiently respond to T cell-mediated help for fungal killing and clearance. We previously showed that microglia, the most numerous population of brain-resident macrophages, were unable to provide protection against *C. neoformans* infection, in either an intravenous acute infection model[5], or in a chronic intranasal infection model[6]. Instead, we found that microglia could host viable *C. neoformans* and may help protect yeast against nutrient starvation in this tissue[5]. Indeed, other studies in various *C. neoformans* infection models using mice, zebrafish, and organotypic brain slice cultures have also shown *C. neoformans* uptake and/or survival within microglia[38–40]. In the current work, although we found evidence that a small population of microglia appeared to respond to IFNγ made by infiltrating CD4 T cells (i.e. IAMs), these cells remained poorly fungicidal. IAMs expressed many genes that are directly

**Fig. 8 | IAMs are a proliferating microglia subset. a** Pie chart showing proportion of cell-cycle genes expressed by IAMs that are highly expressed at specific stages of the cell cycle. Genes were assigned to specific stages of the cell cycle using annotations published in reference[35]. See Supplemental Data 2 for a full list of genes included in the analysis and their cell cycle stage assignment. **b** Example flow cytometry plots of Ki67 staining within total microglia, and staining for IAMs markers CD206 and MSR1 in the Ki67$^-$ and Ki67$^+$ microglia populations. **c** Graph of frequency of total Ki67$^+$ microglia at various time points post-infection (day 0, $n = 7$ mice; day 3, $n = 3$ mice; day 6, $n = 6$ mice; day 9, $n = 5$ mice). Data shown as mean +/- SEM and pooled from 2 independent experiments, with $n = 5–8$ mice analysed per time point. **d** Graph of frequency of Ki67$^+$ positive cells within MSR1$^+$CD206$^+$ microglia (IAMs; filled bars) and MSR1$^-$CD206$^-$ microglia (non-IAMs; open bars) at day 6 ($n = 8$ mice) and day 8 ($n = 5$ mice) post-infection. Data pooled from 2 independent experiments, with $n = 8$ (day 6 post-infection) or 5 (day 8 post-infection) mice analysed. Whiskers refer to the maximum/minimum values, the box refers to interquartile ranges, the centre line refers to the mean. Data analysed by two-way ANOVA, ****$P < 0.0001$, *$P = 0.0103$. **e** Example confocal images of Ki67 labelling of sections from uninfected or infected (day 8 post-infection) brains. An area of fungal infection is outlined with dashed white lines. White arrows denote Ki67$^+$ proliferating cells. **f** Quantification of Ki67 staining relative to DAPI. Pixel density of each stain was calculated using ImageJ relative to section size. Each point represents a section analysed from individual mice ($n = 2$ mice analysed per group). **g** Brain and **h** lung fungal burdens from wild-type mice infected intranasally with *C. neoformans* at day 14 ($n = 12$ mice) and day 21 ($n = 12$ mice) post-infection. Data analysed by Mann Whitney U-test, **$P = 0.0088$. **i** Number of IAMs and (**k**) Ki67$^+$ microglia in brains of wild-type mice infected intranasally with *C. neoformans* at day 14 ($n = 12$ mice) and day 21 ($n = 12$ mice) post-infection. **j** Frequency of Ki67$^+$ cells within the IAMs and non-IAMs populations at day 14 post-infection ($n = 12$ mice). Data analysed by paired t-test, ****$P < 0.0001$. Each point represents an individual mouse, with lines denoting matched samples from the same animal. Intranasal data pooled from two independent experiments and analysed by unpaired t-tests. In all box-and-whisker plots, whiskers refer to the maximum/minimum values, the box refers to interquartile ranges, the centre line refers to the mean.

regulated by IFNγ stimulation, yet harboured more live fungus than other microglia that had less expression of these genes. We believe this supports a model by which CD4 T cells target fungal-infected myeloid cells and deliver IFNγ to them to activate fungal killing pathways and limit infection. Importantly, we used a highly virulent strain for these studies, H99, which typically has higher brain invasion capacity than other *C. neoformans* strains[5]. This pathogenic potential of strain H99 may therefore have exaggerated our results examining fungicidal capacity of IAMs. However, we were able to demonstrate a similar lymphocyte-dependent inflammatory microglia proliferation using an unrelated *C. neoformans* strain (Zc1) and found that IAMs generated in response to *B. dermatiditis* infection, indicating that IAM generation may broadly occur during fungal brain infection.

In an acute infection, signals other than IFNγ may be needed to boost fungicidal activity of microglia, or stimulation of other myeloid cells in the brain (such as monocyte-derived inflammatory macrophages) may be sufficient for protection. Indeed, studies of parasite infection in the CNS have demonstrated divergent responses between brain-resident microglia, border macrophages and inflammatory macrophages. Following trypanosome infection, microglia were responsible for early defence and helped recruit monocytes into the brain that differentiated into inflammatory macrophages, eventually outnumbering resident cells. After infection, microglia returned to their pre-infection transcriptional state, whereas border macrophages retained a long-term genetic signature after infection[41]. In contrast, during *Toxoplasma gondii* infection, microglia released IL-1α, which was required for parasite control, whereas border macrophages produced IL-1β and drove inflammation during infection[42]. Future work should examine the influence of IFNγ and/or CD4 T cells on other types of brain myeloid cells, such as border macrophages, to determine if these other populations have more efficient fungal killing capacity when stimulated by IFNγ.

The inflammatory microglia identified in our dataset appeared to resemble another subset that had been described in an earlier study examining LPS-associated brain inflammation[12]. We named these microglia IAMs, following the naming convention used in the original study[12]. We show here that one of the major markers for IAMs was the scavenger receptor MSR1, which was recently found to be an important phagocytic receptor for non-opsonised *C. neoformans* yeast[30]. Upstream signals that activate microglia towards the IAMs phenotype will be an important future direction to determine how to target these cells in diseases where they play a key role in pathogenesis. Indeed, we found that IAMs significantly expanded after becoming highly proliferative following infiltration of the brain by CD4 T cells, which significantly contributed towards inflammation in the brain, similar to what has been observed in animal models of multiple sclerosis[34]. Targeting microglia proliferation in this disease using the antiviral drug

ganciclovir helped dampen inflammation, which was attributed to a reduction in the recruitment of autoreactive CD4 T cells[34]. However, our data indicate that microglia proliferation was largely dependent on the presence of lymphocytes, since microglia did not proliferate during infection in mice lacking these cells. Other studies have shown that ganciclovir does not directly affect microglia proliferation[43] but can directly affect T cell phenotype and function[44]. The mechanisms by which ganciclovir treatment affect microglia-dependent inflammation within the brain is therefore unclear, but likely affects the interaction between microglia and brain-infiltrating T cells, which together drive inflammation in this tissue.

The observation that brain-infiltrating CD4 T cells drive microgliosis during *C. neoformans* infection may help shed light on the initial events that contribute towards the development of inflammatory syndromes observed in some patients[10]. These inflammatory disorders, broadly termed IRIS, may present as 'unmasking IRIS' where there is no history of cryptococcal meningitis and patients develop infection upon regaining immune function during ART initiation[36]. Alternatively, patients may develop 'paradoxical IRIS', in which an existing *C. neoformans* infection worsens with ART and increasing CD4 counts[36]. In our work, we found that brain infiltration of CD4 T cells appears to be restricted, with antigen-specific populations only entering the CNS in measurable numbers after high fungal burdens had been established. This in turn coincided with the expansion of IAMs, which were present in uninfected brains and not exclusive to the *C. neoformans* infected brain but rapidly increased in number after CD4 T cell recruitment. Our work indicates that proliferating microglia may be an important aspect of IRIS pathogenesis. Future work should explore whether similar events occur in the human brain infected with *C. neoformans*, which we have not been able to demonstrate in the current study. Indeed, multiple mouse models of IRIS have been developed to better understand the pathogenesis of this condition. The majority use a serotype D strain of *C. neoformans* (52D) which causes a chronic brain infection that is cleared by the immune system but triggers an excessive inflammatory response mediated by infiltrating CD4 T cells[45,46]. Blocking entry of these CD4 T cells can alleviate clinical symptoms of the infection, without affecting control of fungal brain infection[45]. These models depend on less virulent strains, which can establish chronic brain infection without mortality, yet they cause human disease less commonly than serotype A strains[47]. Other groups have therefore sought to develop mouse models of IRIS with more virulent serotype A strains, such as we have used here. In those models, lymphocyte-deficient hosts (*Rag*-deficient) are given an adoptive transfer of CD4 T cells, and subsequently develop an inflammatory meningitis upon *C. neoformans* infection that is characterised by upregulation of aquaporin-4 and dysregulation of water flux in the brain[48]. These models are dependent on using lymphocyte-deficient

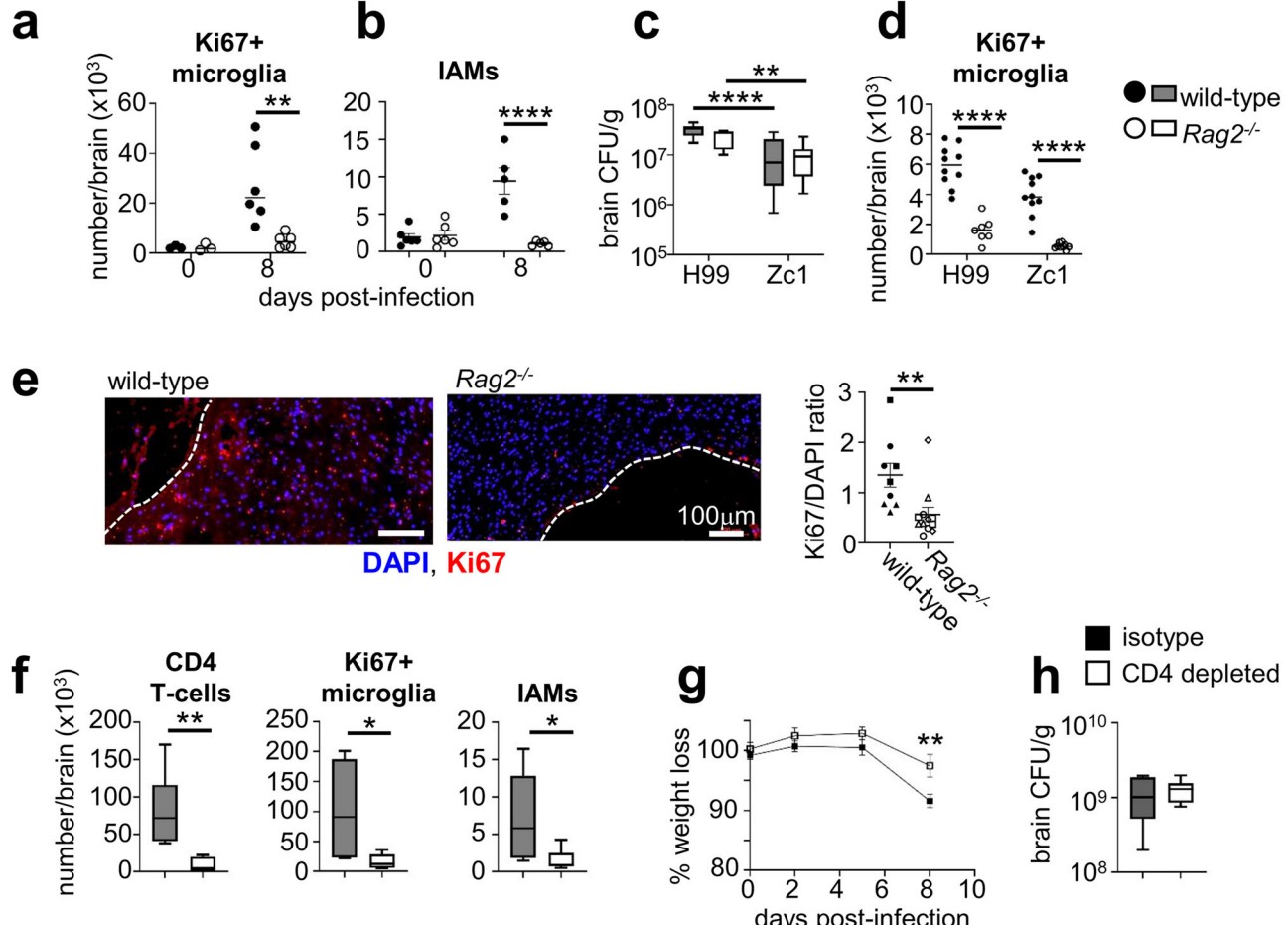

**Fig. 9 | Brain-infiltrating CD4 T cells drive microglia proliferation during infection. a** Number of Ki67⁺ microglia and **b** IAMs (CD206⁺MSR1⁺ microglia) at day 0 or day 8 post-infection in wild-type (n = 6 mice; filled symbols/bars) and *Rag2⁻/⁻* (n = 6 mice; open symbols/bars) mice. Data pooled from two independent experiments, presented as mean +/- SEM and analysed by two-way ANOVA. **P = 0.002, ****P < 0.0001. **c** Fungal brain burdens in wild-type and *Rag2⁻/⁻* mice at day 8 post-infection in mice infected with *C. neoformans* strain H99 (n = 10 wild-type mice, n = 7 Rag2-deficient mice) or *C. neoformans* strain Zc1 (n = 10 wild-type mice, n = 9 Rag2-deficient mice), and **d** number of Ki67+ microglia at day 8 post-infection in mice infected with *C. neoformans* strain H99 (n = 10 wild-type mice, n = 7 Rag2-deficient mice) or *C. neoformans* strain Zc1 (n = 10 wild-type mice, n = 9 Rag2-deficient mice). Data pooled from two independent experiments and analysed by two-way ANOVA. **P = 0.0052, ***P < 0.0001. **e** Example immunofluorescence of Ki67 and DAPI labelling in brain sections from wild-type and *Rag2⁻/⁻* mice at day 8 post-infection. Infection lesions are outlined with white dashed lines. Images were quantified by randomly selecting three parenchymal regions near sites of infection within each brain section, and dividing the pixel intensity of the Ki67 stain by the pixel density of the DAPI stain

for each analysed region to determine the Ki67/DAPI ratio. Three mice were analysed for wild-type mice (9 regions sampled) and 4 *Rag2⁻/⁻* mice analysed (12 regions sampled). Different symbol shapes show matched regions for each mouse. Data analysed by unpaired t-test, presented as mean +/- SEM. **P < 0.008. **f** Total number of CD4 T cells, Ki67⁺ microglia and IAMs in the brains of mice at day 8 post-infection, treated with either isotype control antibody (n = 6 mice; filled symbols/bars) or anti-CD4 depleting antibody (n = 6 mice; open symbols/bars). Data pooled from two independent experiments and analysed by unpaired t-tests. *P = 0.0447 (IAMs), *P = 0.0364 (Ki67+ microglia), **P = 0.0059. **g** Weight loss, relative to weight at start of experiment, of isotype control or CD4-depleted mice (n = 9 per group) during infection. Data shown as mean +/- SEM, pooled from three independent experiments. Data analysed by two-way ANOVA with Bonferroni correction, **P = 0.0045. **h** Fungal brain burdens at day 8 post-infection in isotype or CD4-depleted mice (n = 6 mice per group). Data pooled from two independent experiments. In all box-and-whisker plots, whiskers refer to the maximum/minimum values, the box refers to interquartile ranges, the centre line refers to the mean.

---

mouse strains. However, recent work has demonstrated that microglia are not fully mature in these mouse strains[49], further underscoring the influence of CD4 T cells on microglia behaviour and function that remains incompletely understood.

In summary, our data identifies CD4 T cell-dependent microgliosis as an important component of brain inflammation that accompanies *C. neoformans* infection. We demonstrate functional roles for the IAMs activation phenotype of microglia, diversifying our understanding of the range of functional phenotypes that microglia may adopt under different inflammatory conditions. Collectively, these data improve our understanding of how inflammation develops in the brain during this life-threatening fungal infection and identifies several potential targets for future therapeutic intervention.

## Methods

### Mice

8–12 week old mice (males and females) were housed in individually ventilated cages under specific pathogen free conditions at the Biomedical Services Unit at the University of Birmingham, and had access to standard chow and drinking water *ad libitum*. Mice were housed under 12 h light/dark cycle at 20–24 °C and 45–65% humidity. Experiments with transgenic mice utilised both males and females to maintain littermate controls. We used female mice for experiments with wild-type mice only, since female mice can be housed in larger groups/smaller cage numbers. Wild-type refers to C57BL/6NCrl (Charles River) or the corresponding littermates of genetically-modified lines; *Sall1-Cre^ER Rosa26^Ai14* and *Rag2⁻/⁻*. *Rosa26^Ai14* mice (all on C57BL/6 J

background) were originally purchased from Jackson and colonies bred and maintained at the University of Birmingham. *Sall1*-Cre[ER] mice were a kind gift from Dr Melanie Greter (University of Zurich). For adoptive transfer experiments, CnT.II mice (on C57BL/6 background)[18] (a gift from Prof. Kazuyoshi Kawakami, University of Tohoku) were used as donors. Alternatively, CnT.II mice were crossed with Nr4a3-Tocky[20] animals that had been crossed with Great-SMART[50] cytokine reporter animals (both on C57BL/6 background). An F1 cross between CnT.II and Nr4a3-Tocky-Great-SMART reporters were used for donors in adoptive transfers. Nr4a3-Tocky mice were developed in Imperial College, London (Ono lab)[20]. Colonies of CnT.II and Nr4a3-Tocky-Great-SMART were maintained at the Biomedical Service Unit as above. In all experiments, mice were euthanised by cervical dislocation at indicated analysis time-points, or when humane endpoints (e.g. 18-20% weight loss, hypothermia, meningitis) had been reached, whichever occurred earlier.

### *C. neoformans* growth and mouse infections
*C. neoformans* strains used in this study were H99, Zc1[37], and KN99a-GFP[51]. Yeast was routinely grown in YPD broth (2% peptone [Fisher Scientific], 2% glucose [Fisher Scientific], and 1% yeast extract [Sigma]) at 30 °C for 24 h at 200 rpm. For infections, yeast cells were washed twice in sterile PBS, counted using haemocytometer, and $2 \times 10^4$ yeast injected intravenously into the lateral tail vein. In some experiments, mice were infected intranasally with $2 \times 10^5$ yeast under isoflurane anaesthesia. For analysis of brain and lung fungal burdens, animals were euthanized and organs weighed, homogenized in PBS, and serially diluted before plating onto YPD agar supplemented with Penicillin/Streptomycin (Invitrogen). Colonies were counted after incubation at 37 °C for 48 h.

### *Blastomyces dermatitidis* infections
C57BL/6 mice were intravenously infected with 100,000 BAD1 null yeast of *Blastomyces dermatitidis*[52]. Animals that showed neurological signs of fungal infection (head tilt, spinning, paralysis etc.) were euthanised and the brain harvested. Brain tissue was homogenized with the gentleMACS using program mlung1.1 and then digested with 1 mg/ml collagenase D and 20 μg/ml DNAse I in RPMI at 37 °C under 200 rpm agitation for 25 min. The digested brains were ground again in the gentleMACS using program mlung2.1 and the enzymatic reaction stopped with 100 μl of 0.5 M EDTA. An aliquot of the digested brain was plated for CFU on BHI agar plates. After pelleting the homogenate, the pellet was resuspended in 8 ml of 40% Percoll and underlaid with 80% Percoll. After spinning the Percoll gradient at 1578 g for 20 min at room temperature the interface was collected and filtered with DYU Nylon filter and washed with 7 ml RPMI. The whole pellet was stained and analysed by flow cytometry.

### Brain disassociation and cell isolation
Brains were aseptically removed and immediately stored in ice-cold MACS Tissue Storage Buffer (Miltenyi). Brains were then rinsed in ice-cold Dulbecco's PBS (D-PBS) and finely minced with a scalpel. Brain tissue was then digested using the MACS Adult Brain Disassociation Kit (Miltenyi), using the 37C_ABDK_01 program on the gentleMACS tissue disassociator following the manufacturers' instructions. Digested brains were filtered using a 70 μm filter, washing through with ice cold D-PBS, followed by centrifugation at 400 g at 4 °C for 5 min. The cell pellet was resuspended in 10 mL 30% Percoll (GE Healthcare), diluted in sterile PBS supplemented with 10% FBS. The suspension was underlaid with 2 mL 70% Percoll, and the gradients spun at 1000 g for 30 min at 4 °C, with the brake off. Cells at the interphase were collected and washed in sorting buffer (sterile PBS supplemented with 0.5% FBS and 2 mM EDTA) prior to staining or cell sorting.

### Cell sorting
Leukocytes isolated from the brain were resuspended in sterile PBS prior to staining with ZombieViolet viability dye (Biolegend; 1:200) for

10 min on ice. Fc receptors were blocked using anti-CD16/32 (Biolegend; 1 μg/mL) and cells were subsequently stained with anti-CD45-PE and anti-CD11b-APC Cy7 (Biolegend; 1 μg/mL) for 20 min on ice. Cells were washed in sorting buffer (PBS + 0.5% FBS + 2 mM EDTA) and immediately sorted on a BD FACS Aria (BD Biosciences).

### Single Cell RNA sequencing
**Sorting and RNA isolation.** Myeloid cells from the brain were sorted as live CD45 + CD11b+ singlets, and collected into RPMI media supplemented with 10% FBS and 1% Pen/Strep, keeping cells cold during the sort. Sorted cells were spun down and resuspended into PBS supplemented with 0.04% BSA at a concentration of approximately 1000 cells/uL. Cells were loaded on a GemCode Single-Cell Instrument (10X Genomics) to generate single-cell gel bead-in emulsions (GEMs). scRNAseq libraries were prepared using a Chromium Next GEM Single-Cell 3' Library and Gel Bead Kit v2 (day 0/3 samples) or v3.1 (day 6 sample) (10X Genomics).

**RNA Seq analysis.** Sequencing libraries were loaded on an Illumina NextSeq500 with the following settings; 1.6pM initial loading concentration, 28 cycles read 1, 98 cycles read 2, 8 cycles index, v2.5 150 cycle flow cell. Sequencing was performed at Genomics Birmingham (University of Birmingham, UK). The demultiplexing of raw sequencing data was performed using CellRanger v3.0.1 (10X Genomics), using function mkfastq. Reads obtained from mkfastq were used as the input for 'cellranger count' (10X Genomics), in which reads were aligned to the mouse reference genome (mm10, v3.0.0; Ensembl 93). Pooling of the different samples was completed using 'cellranger aggr' (10X Genomics), in which the different chemistries used for different samples was accounted for and corrected. Raw sequencing data and processed data files have been deposited in the GEO database under accession number GSE262502.

**Data pre-processing and filtering.** Sequenced data processed by CellRanger was filtered, checked and analysed using Partek Flow (v9.0.20). H5 files generated by CellRanger were imported into Partek software which uses a Seurat-based pipeline for analysis. We applied the following filters during analysis: total reads per cell 500-12,000, number of expressed genes 250–3000, percentage of mitochondrial reads maximum 8%. These filters retained 99.18% of cells (15,398 out of 15,526). Normalisation was then performed, using the recommended settings (counts per million, +1, log2 transform). Then, a noise reduction filter was applied and any genes not expressed by any cell in the data set were filtered out (of 31,017 genes, 12,460 genes were removed). Since our samples were run on different time points, we performed correction steps to remove batch effects. For this, we performed a clustering analysis and visualised the results by tSNE (using the following parameters: Louvain clustering, 30 nearest neighbours [K-NN], 10 principal components). We classified cells expressing high levels of *Cx3cr1, Sall1* and *Tmem119* as 'microglia', and cells expressing high levels of *H2-Aa, Cd163* and *F13a1* as 'macrophages'. We then used the 'Remove Batch Effect' function, removing the attribute 'time-point' and the interaction between this attribute and the 'classifications' applied previously. We then re-ran the clustering analysis and analysed by tSNE to evaluate the removal of the batch effect. Graph-based clustering, detailed cellular classifications and downstream analyses (pathway analysis, differential gene expression) were performed on this clustering result.

### Flow cytometry
Isolated leukocytes were resuspended in PBS and stained with Live/Dead stain (Invitrogen) on ice as per manufacturer's instructions. Fc receptors were blocked with anti-CD16/32 and staining with fluorochrome-labelled antibodies was performed on ice. Labelled samples were acquired immediately or fixed in 2% paraformaldehyde

prior to acquisition. In some experiments, labelled samples were fixed and permeabilised for staining for intracellular antigens (Ki67, arginase). Fixing and permeabilisation was performed using the Foxp3 Staining Buffer Set (eBioscience) as per the manufacturers instructions, and samples stained for intracellular antigens overnight at 4 °C in permeabilisation buffer prior to washing and acquisition the following day. Anti-mouse antibodies used in this study were: CD45 (30-F11), CD11b (M1/70), CX3CR1 (SA011F11), MHC Class II (M5/114.15.2), F480 (BM8), Ly6G (1A8), Ly6C (HK1.4), CD44 (IM7), CD69 (H1.2F3), CD127 (A7R34), CD45.2 (A20), CD45.1 (104), TCRβ (H57-597), CD38 (90), CD206 (C068C2), γδ TCR (GL3), Ki67 (16A8) all Biolegend, and CD4 (RM4.5), CD62L (MEL-14) from BD Biosciences, and MSR1 (M204PA), arginase (A1exF5) from Invitrogen. All antibodies were used at a final concentration of 1 μg/mL, except for anti-CD206 and anti-F4/80 (2 μg/mL) and anti-MSR1 (4 μg/mL). Samples were acquired on a BD LSR Fortessa equipped with BD FACSDiva v9.0 software. Analysis was performed using FlowJo (v10.6.1, TreeStar).

## MHCII tetramer staining

Leukocytes were isolated from brains as described above. Cells were stained with 9.83 nM APC-labelled MHCII tetramer (I-Ab) loaded with Cda2 peptide (sequence HQYMTALSNEVVF) in FACS buffer for 45 minutes in the dark at room temperature. Cells were then washed as above before staining for surface markers and flow cytometry analysis as described above. MHCII Tetramers were generated by the NIH Tetramer Core Facility at Emory University, USA.

## Sorting of brain immune cells and ex vivo killing assay

Mice were infected with GFP-expressing *C. neoformans* as above, and brain leukocytes isolated at day 7 post-infection. Infected (GFP + ) IAMs microglia (CD45intCX3CR1hiCD11b+CD206+MSR1+), and non-IAMs microglia (CD45intCX3CR1hiCD11b+CD206−MSR1−) were FACS purified using a BD FACS Aria Fusion cytometer under sterile conditions. Collected cells (1000-2000 total) were centrifuged, the pellet resuspended in 1 mL sterile tissue-culture grade water, and samples were then plated onto YPD agar plates (200-300μL per plate). Plates were incubated at 30 °C for 2 days prior to colony counting, and killing rate calculated as a percentage of colonies counted over total number of infected cells plated.

## ImageStream analysis

Infected IAMs and non-IAM microglia were FACS purified as above and then fixed in 2% paraformaldehyde. Fixed samples were imaged using an Amnis ImageStream X MkII equipped with three lasers and 6 optical detectors. Final analysis was completed using IDEAS software (version 6.2).

## Adoptive transfers

CD4 T cells were isolated from lymph nodes and spleens of CnT.II or CnT.II-Nr4a3-Tocky mice using magnetic beads-based CD4 T cells isolation kit (Miltenyl Biotec) following the manufacturer's instruction. Purified CD4 T cells were stained with 10 μM cell proliferation dye CFSE (Biolegend) or CellTrace eFluor450 (eBioscience) according to manufacturer's instructions. $5 \times 10^6$ CD4 T cells cells were injected into recipient mice intraperitoneally. Mice were then infected with *C. neoformans* as above ~24 h after adoptive transfer. Mice were sacrificed on day 7 post-infection, and brains, lungs and spleen were dissected for further analysis by flow cytometry as above. In some experiments, recipient mice were injected with 80 or 200 μg Cda2 peptide (HQYMTALSNEVVF, generated by GLS, China) intraperitoneally on day 7 post-infection. Mice were sacrificed 4 h after peptide injection, and the brains and spleens were dissected for further analysis.

## CD4 T cell depletions

Wild-type C57BL/6 female mice were injected intraperitoneally with 300 μg anti-mouse CD4 antibody or 300 μg IgG2b isotype control

(both BioXCell) on day −2, day 0 and day 5 relative to infection. Depletion of CD4 T cells was confirmed by flow cytometry in the peripheral blood and brains at time of analysis.

## Brain immunofluorescence

Whole mouse brains were fixed in 4% paraformaldehyde for 24 h at 4 °C then placed in 30% sucrose solution at 4 °C for 24 h (or until the brain sank), before freezing in OCT (Epredia). Brains were sectioned horizontally (at 30 μm) using a cryostat (Bright Instruments). Slides were washed in ice-cold PBS prior to blocking for 1 h at room temperature in blocking solution (0.1 M Tris, 1% BSA, 0.1% donkey serum, 0.05% Triton X-100, 0.01% Saponin). Brains were stained with anti-Iba-1 antibody (1:500, Wako) performed in blocking solution overnight at 4 °C, followed by Alexa-Fluor 647 secondary antibody (1:500, Thermofisher) incubated for 2 h at room temperature and nuclei were stained with DAPI (1:1000). In Ki67 labelling experiments, brains were fixed in 10% formalin for 24 h prior to embedding in paraffin. Formalin fixed paraffin embedded (FFPE) brain tissue sections (5 μm) were dewaxed and rehydrated using standard protocols. Antigen retrieval was performed in 10 mM citric acid monohydrate (pH 6) at 95 °C for 10 min. Endogenous fluorescence was quenched with 20 mM ammonium chloride. Non-specific binding was blocked for 1 h in a buffer containing 10% normal goat serum, 3% bovine serum albumin and 0.05% Tween 20 in PBS. A rabbit monoclonal anti-Ki67 antibody (Abcam, 10 μg/ml) was applied overnight at 4 °C in blocking buffer. Goat anti rabbit IgG conjugated to Alexa Fluor 647 (Invitrogen, 1:1000) was incubated for 1 h at room temperature. Tissue sections were treated for autofluorescence quenching with the TrueVIEW Autofluorescence Quenching Kit (VectorShield) prior to mounting the slides with ProLong Gold Antifade mountant (Thermofisher). The slides were imaged using the Zeiss Axio Scan Z1 slide scanner and the Zeiss LSM 880 Confocal with Airyscan Fast. Image analysis was performed using the Zen Blue (v3.1) software and ImageJ.

## Quantitative PCR

FACS purified cells were pelleted by centrifugation and resuspended in 50 μL Trizol reagent (Thermo) and transferred to RNAse-free eppendorfs and increasing the volume of Trizol to 1 mL final. Samples were frozen at −80 °C prior to RNA isolation. For that, samples were thawed at room temperature and 200 μL cholorform added. Samples were centrifuged at $13,000 \times g$ for 15 min at 4 °C. The upper phase was transferred to a new RNAse-free Eppendorf and then mixed with 500 μL isopropanol and 5 μL GlyoBlue co-precipitant (Thermo) and stored at −20 °C overnight. The next morning, RNA was pelleted by centrifugation ($13,000 \times g$ for 5 min at 4 °C) and then washed in 700 μL ice-cold 70% ethanol. RNA pellets were air-dried and resuspended in 10-20 μL nuclease-free water. RNA was converted to cDNA using the High Capacity cDNA kit (Thermo) as per manufacturer's instructions. cDNA was used as a template for qRT-PCR, using relative quantification method and SYBR green detection. Briefly, master mixes were made using SYBR Green PowerUp (Thermo) and primers at final concentration of 300 nM. qRT-PCR reactions were performed using a QuantStudio5 machine (Thermo) and analysis performed using ThermoFisher Connect software. Primer sequences were as follows: *Tmem119* (Fwd: AATGACAGCTCTTCACCGGG, Rev: GCATGCACCGC-TATATTGGC), *RPL13a* (Fwd: GCGGATGAATAC CAACCCCT, Rev: CCACCATCCGCTTTTTCTTGT).

## In vitro T cell stimulations

Splenocytes were isolated from C57BL/6 mice and seeded at $5 \times 10^5$ cells per well in 96 well U-bottom plate, and loaded with 3 μg/mL Cda2 peptide (HQYMTALSNEVVF) generated by GLS (China). CD4 T cells were then isolated from lymph nodes and spleen of CnT.II mice using MACS CD4 T Cell Isolation (Miltenyi Biotec) according to manufacturer's instruction. Purified CD4 T cells were stained with 1 μM CFSE

(Biolegend) for 5 minutes at room temperature and washed twice in RPMI supplemented with 10% FBS. $1 \times 10^5$ CFSE-stained CD4 CnT.II cells were co-cultured with the peptide-loaded splenocytes. In some experiments, isolated CnT.II T cells were cultured in anti-CD3/28-coated plates. Plates were coated with purified functional grade anti-CD3 (1452C11; 2 µg/ml; eBioscience) and anti-CD28 (37.51; 10 µg/ml; eBioscience). Cells were incubated for 4 h and then collected for flow cytometry using a pipette, staining and analysing as above.

## Statistics

Statistical analyses were performed using GraphPad Prism 10.0 software. Details of individual tests are included in the figure legends. In general, data were tested for normal distribution by Kolmogorov-Smirnov normality test and analyzed accordingly by unpaired two-tailed *t*-test or Mann Whitney *U*-test. In cases where multiple data sets were analyzed, two-way ANOVA was used with Bonferroni correction. In all cases, *P* values < 0.05 were considered significant.

## Ethics statement

All research complied with local ethical approval. All animal studies and associated ethics were approved by the Animal Welfare and Ethical Review Board at the University of Birmingham and under project licences PBE275C33 and PP7564605.

## Reporting summary

Further information on research design is available in the Nature Portfolio Reporting Summary linked to this article.

## Data availability

All raw data associated with the figures are without restriction. Raw sequencing data and processed data files have been deposited in the GEO database under accession number GSE262502. All other data are available in the article and its Supplementary files or from the corresponding author upon request. Source data are provided with this paper.

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

## Acknowledgements

We would like to thank the technical staff at the Biomedical Services Unit (Birmingham) for their care and help with animal husbandry. In particular, we thank Claire Lyons and Karen Boswell for their time and patience performing infections and caring for our animals. We thank Dr Ferdus Sheik and support staff at the Technology Hub and Flow Cytometry Unit at the University of Birmingham for their support with sorting, flow cytometry and microscopy experiments. This work was funded by: Academy of Medical Sciences (SBF004_1008, awarded to RAD), Medical Research Council (MR/S024611 awarded to RAD and MR/T029137 awarded to RAD and KK), Wellcome Trust Institutional Strategic Fund (University of Birmingham, awarded to RAD), Wellcome Trust PhD studentship (awarded to SAH), Hartwell Foundation Fellowship (awarded to AJW), Biotechnology and Biological Sciences Research Council (BBSRC) David Phillips Fellowship (BB/J013951, awarded to MO), British Heart Foundation BHF Intermediate Fellowship (FS/IBSRF/20/25039, awarded to JR).

## Author contributions

S.H., M.S.F., L.W., L.G., D.L., A.J.W., E.C., J.R. and R.A.D. performed the experimental studies. S.H., M.S.F., A.J.W., M.W., J.R., D.B. and R.A.D. carried out the analysis. M.S.F., M.W., B.K., K.K., J.R. and R.A.D. supervised the work. R.A.D., M.O., B.K. and K.K. obtained the funding. K.O., M.W., B.K., K.K., D.B. and R.A.D. conceived and designed experiments. M.O., K.O. and K.K. provided key reagents and expertise. S.H., M.S.F. and R.A.D. wrote the manuscript.

## Competing interests

The authors declare no competing interests.
