## [Transparent Peer Review file · Nature Communications]

Brain-infiltrating CD4 T-cells drive inflammatory microglia proliferation during cryptococcal meningitis

Corresponding Author: Professor Rebecca Drummond

Version 0:

Reviewer comments:

Reviewer #1

(Remarks to the Author)

In their manuscript "Brain-infiltrating CD4 T-cells drive inflammatory microglia proliferation during cryptococcal meningitis", Hain et al. provide a detailed characterization of the immune response to brain infection with *C. neoformans*, focusing on the interplay between CD4 T helper cells and myeloid cells, including microglia. Using a combination of flow cytometry, microscopy, and single-cell RNA sequencing (scRNA-seq), alongside TCR transgenic mice and genetic reporter and fate-mapping models, the authors offer valuable insights into T cell reactivity and myeloid cell identity. The manuscript is well-written, with a particularly clear and engaging introduction. However, the sequencing analysis is relatively weak and could benefit from employing the Seurat pipeline for more robust interpretation. Additionally, the flow cytometry characterization of 'inflammation-associated microglia' (IAMs) appears to be affected by incorrect gating, and clarification is needed to distinguish BAMs from microglia. This could be addressed by leveraging the authors' Sall1 fate-mapping strain and incorporating immunofluorescence to differentiate CNS border and parenchymal populations. The discussion could be shortened. A comparison with work from the Rua or Movahedi labs on brain macrophages in infection would add important context. While there are areas for improvement, this study represents a valuable contribution to our understanding of immune responses during fungal CNS infections.

Specific:

Figure 1:

- 1f: A longitudinal analysis with quantification would help clarify the infiltration patterns of fungus-specific T cells. It seems unlikely that T cells initially infiltrate via brain capillaries across the BBB unless significant BBB damage occurs early on. Do the authors observe (fungus specific) T cells in the entry sites (leptomeninges and choroid plexus) between day 6-8? Could the authors do in vitro fluorescent labeling of TCRtg cells (alternatively, use the congenic marker as in Fig 2) and transfer them to infected mice to track them? A quantification of the imaging part(s) would be appreciated.

Figure 2:

- c-d: Could the authors provide raw plots to show how they gated on CD44+ and proliferating T cells? Also it would be nice to look at those cells in the cLNs and get more information on the kinetics. As there is only one mouse analyzed on day 4 in the lung in c and d (but 2 in b?) it is not possible to say if the kinetics are more similar to the spleen or more similar to the lung. Typically, priming of fungus-specific T cells occurs in secondary lymphoid organs, with subsequent migration to infected tissues.

Figure 4:

- The experiments with Tocky mice are intriguing. Could the authors show comparisons of brain, lung, and spleen using polyclonal TCR Tocky mice (Fig. 4e) to understand whether differences persist across tissues?

Figure 5:

- Could the authors provide their gating strategy for the cell populations in Fig. S3A?

- The analysis in Fig. S4 and 5 appears somewhat redundant to the flow analysis. Why were the time points split? A more integrated dataset could highlight temporal changes in myeloid cell composition more effectively. Heatmaps/dot plots of major clusters and volcano plots for DEG analysis between time points could simplify interpretation.

- Fig. S6 is not clearly explained in the text. It seems to depict BAM populations over time, but this should be introduced and interpreted in the results. In light of my concerns about the FC part of IAM characterization, a clearer BAM analysis could align better with the manuscript's main message.

- The choice to focus on cluster 10 is unclear. Cluster 10 does not appear unique in the dot plot (e.g., lacking MHCII

expression). Could the authors take an unbiased approach, identifying the microglia cluster most affected (e.g., clusters with the highest DEG activity on days 3 or 6)? This analysis could refine the study's focus.

- Comparing the DEGs with available signatures of isolation-induced transcript is commendable, but it is not relevant to their question as these signatures should be induced at all time points and, therefore, not contribute to DEGs?

Figure S8:

- Line 246: Could the authors clarify how the genes in cluster 10 were identified? This information is missing.

- Line 248: The authors state they aim to examine how CD4 T cells influence cluster 10. However, as cluster 10 only appears on day 6, this question is challenging to address. Comparing microglia at day 3 (pre-T cell influx) with day 6 could be more informative. The introduction to this analysis might be better placed in a later section (e.g., Figure 8).

- Panels S8A–B: The analysis here is somewhat unconventional. Could the authors compare the top 10–20 DEGs between DAMs and IAMs? More clarity is needed on what makes cluster 10 distinct

Figure 6:

- Line 281: The claim that IAMs express Mrc1 (CD206) is not apparent from the cluster description in Fig. S3.

- The flow cytometry gating in Fig. 6B likely captures BAMs, not microglia (BAMs typically express CD204 and CD206 at high levels, while microglia do not). To confirm microglia identity, the authors should perform immunofluorescence (IF) to localize IAMs in the parenchyma and demonstrate Sall1 expression in their IAM gate. Raw flow plots supporting Fig. 6f,g would be helpful.

- Given the rarity of cluster 10 (<10%), preselecting for microglia using Sall1 expression might be necessary before identifying CD204+ subsets. Could the authors clarify whether cluster 10 is defined by CD206 expression in their scRNA-seq data?

Figure 7:

- The gating strategy here should be included in the supplement. It appears the data may reflect BAMs rather than microglia. Repeating the analysis using Sall1 fate-mapping mice and a more precise gating strategy would help.

- Please include the gating on GFP+ IAMs and non IAMs

- The "microglia-depleted brain" image is confusing, as this analysis is not described in the text. Is this a leftover of their previous publication? <https://www.nature.com/articles/s41467-023-43061-0#MOESM1>. If relevant, it should be clarified.

- Using the Sall1 fate-mapping strain to visualize fungal uptake would enhance this figure. Including comparisons at brain borders (e.g., CD204+/CD206+ BAMs) versus parenchyma would be valuable. Quantification of these images is also recommended.

- The references to panels 7f and 7g seem incorrect; they likely refer to panels 7g and 7h, respectively. Could the authors clarify?

- The absence of Arginase induction in microglia from cpl1 mutants is unsurprising. A comparison between microglia and non-microglia populations could support their conclusions.

Figure 8 :

- Please include gating strategies for IAMs in Fig. S9.

- Panels 8b–g seem to analyze BAMs rather than microglia (low CX3CR1 and high CD206 expression). This should either be repeated as described above or focused on BAMs in sequencing and imaging analyses.

- Line 368: The statement on IAM-driven inflammation and its link to CD4 T-cell infiltration is not addressed experimentally

- The role of CD4 T cells is only while the role of IFN γ is not addressed in the manuscript. Consider rephrasing this text to align with the study's flow.

- Could the authors evaluate what happens when microgliosis is blocked, and whether this benefits *C. neoformans* infection and/or the clinical outcome?

- Is the fungal clearance in the brain (and periphery) different in RAGKO mice?

- The title claims that brain infiltrating CD4+ T cells drive microgliosis. Can this also be supported by IF? Do the authors see differences in the depletion of intravascular and brain-invading CD4 cells (using intravascular labeling by CD45 or CFSE as the authors used before)?

Minor:

- L50 abbreviation for iNOS missing

- L213, the authors write they used n=2-5 mice per time point, but the legend says 3-4. What is it? The exact numbers would be more informative as it is only 3 time points.

- L261 type: "RMA" instead of RNA

- L336: NO has been abbreviated before

- In the methods, the authors write about Sall1CreER Csf1r1fl mice, but this strain has not been used

Reviewer #2

(Remarks to the Author)

This is a mouse study describing in great detail the kinetics of the CNS infection with *Cryptococcus neoformans* H99 in a mouse model. This study represents the collaborative work of 3 important fungal immunology groups and used the newest tools to characterize the immune response in this model at 3 distinct time points.

The strengths of this paper are the quality of data and writing and the masterful use of relatively novel tools. The weakness is that this paper represents a purely descriptive work without testing any specific hypothesis. While the authors discuss some possible future directions, such as testing the role of TLR4, no comparisons have been made beyond the detailed model characterization.

There is also a major concern that a single strain of *C. neo* is used (H99), which this group already showed is capable of extensive growth in microglia in contrast to other strains of *C. neo*. *Nat Commun.* 2023 Nov 8;14(1):720

Thus, some of the conclusions about the weak fungicidal capacity of activated microglia could be biased by the choice of

this strain.

Other comments:

The adoptive transfer of TR Tg cells was used to demonstrate that brain recruitment of the antigen-specific T-cells is delayed compared to non-specific T-cells. However, the antigen-specific T-cells are transferred in a low proportion compared to the host T-cells on day -1, and we don't know how long these cells require to populate the lymphatic system and adapt before they assume their physiological functions in vivo. So there is a possibility that the delay is a cell-transfer artifact. Would we obtain a similar outcome if the non-specific cells were transferred to the TCR transgenic mouse?

Some human specimen data validating at least some of the findings presented here would greatly strengthen the paper.

The discussion covers a lot of different topics/areas not tested in this paper.

Reviewer #3

(Remarks to the Author)

In this manuscript by Hain et al, the authors describe the role of inflammatory microglia in a mouse model of cryptococcal meningitis. Although the data are interesting, there are some areas the authors need to address.

Major issues:

1. As the authors mention multiple times in the manuscript, the main population of humans affected with cryptococcal meningitis are AIDS patients, who have an extremely low CD4+ T cell count. Although the data showing CD4+ T cell induction of IAM proliferation (Figure 8) are very interesting, this is somewhat counterintuitive to the situation in AIDS patients, where most symptomatic cryptococcal meningitis is found. Although IRIS is an issue, many clinical studies point to an excessive pro-inflammatory response due to reconstitution. It is difficult, with this study or others, to determine whether or not the reconstitution of CD4+ T cells and their interaction with microglia are specifically involved in IRIS.
2. The intravenous model of infection, while used widely, is not truly representative of trafficking of *C. neoformans* from the lung to the brain to induce meningitis. Since the authors did not compare the brain microglia outcomes from a pulmonary infection to the i.v. infection, it is difficult to assess the actual implications of cell involvement during the natural course of infection, especially when myeloid cells from the lung (which may traffic during a pulmonary infection) are not involved in the i.v. model.
3. While most experiments seem to have replicates, some experiments were only conducted once. Some had different numbers of experiments depending on the time point (for example, Figure 2 shows 1 experiment for D4 vs 2 experiments for D0 and D7), which makes it difficult to assess reproducibility of the results as well as consistency of the experiments (for example, if the D7 and D4 time point were from different adoptive transfers and different fungal inocula). Therefore, it is difficult to assess some of the results (especially those in Figure 2) due to the inconsistency and lack of reproducibility of some of the data shown.
4. In the studies examining IAM fungicidal activity, it is difficult to discern the antifungal activity based on the experimental setup. If the IAM and non-IAM cells contain different numbers of *C. neoformans* cells prior to removal and plating of the cells ex vivo, then it is not possible to calculate killing capacity of those cells. For example, in Figure 7a, the % infected cells is shown, but that does not mean that each cell is only infected with a single cryptococcal cell. Even though the cells are sorted for GFP+ cryptococcal cells prior to plating, that does not mean that those cells contain the same number of GFP+ cells. Therefore, these data are impossible to assay for antifungal activity using these methodologies.

Minor issues:

1. In the legend for Figure 5, the number of animals per group per time point and the number of experiments is missing.

Version 1:

Reviewer comments:

Reviewer #1

(Remarks to the Author)

The manuscript has improved following revision, and most of the points raised during review have been addressed. Thank you for your thoughtful responses and the additional analyses.

Reviewer #2

(Remarks to the Author)

The manuscript has been revised and supplemented with additional data, improving greatly over the previous version. There is one major comment regarding the results linked to Figure 8 and a few minor issues (recommended edits).

Major:

LN386-407; The conclusion that CPL-1, as correctly labeled in methods (elsewhere spelled as CLP-1) promotes arginase

expression in IAM and intracellular residence is not fully supported.

* LN 388: While poor fungal killing was linked to Arg1, causality has never been established here or in other studies. This would require Arg1 inhibitor or Arg1 knockdown to be used, and showed that the cells kill the fungus better when Arg1 is inactive.

* LN 397-402 The scenario presented here is only one of the options. The data shows that the mutant is 2 orders of magnitude lower in the brain; therefore, even if the mutant triggers Arg1, we are not likely to see it due to the sparsity of the fungus (1% of what we see in the WT).

* Quite contrary, the data on Fig 8D shows that despite the very low antigen load, up to 30% of MHCII-hi cells can be Arg1+, and on average their Arg1+ frequency in MHC-hi is similar to that in MHCII-lo cells with the WT, which are only 5% infected.

* We also do not know why there are fewer mutants in the brain: did it not cross the BBB, did it die due to reduced adaptation to the brain environment (such as very low glucose or copper), or other fitness problems?

Therefore, from this data, we cannot conclude that cpl-1 was responsible for Arg1 induction, and this, in turn, was the cause of its lower CFU.

This part is not essential for the flow of the manuscript, while the data is insufficient to sustain the author's conclusion without much additional work to clarify all the issues above. Thus, most likely, we need this part to be removed unless additional convincing data is included.

Minor

LN 50-54 The cited study "2" is a review and original articles should be cited; also, while the information about IFN-g is correct, the role of NO in fungal killing is contextual. Multiple studies showed it to be dispensable or less important than other fungicidal macrophage mechanisms. eg: PMID: 11092381, 11907100, 38841113. Thus, this statement about NO playing a major role should be softened or removed.

LN 147 "proliferated" fits better than "proliferating" here, since we don't know at which site proliferation occurred based solely on low CFSE.

Reviewer #3

(Remarks to the Author)

The updated manuscript has addressed all review criteria.

Reviewer #1 (Remarks to the Author)

In their manuscript "Brain-infiltrating CD4 T-cells drive inflammatory microglia proliferation during cryptococcal meningitis", Hain et al. provide a detailed characterization of the immune response to brain infection with *C. neoformans*, focusing on the interplay between CD4 T helper cells and myeloid cells, including microglia. Using a combination of flow cytometry, microscopy, and single-cell RNA sequencing (scRNA-seq), alongside TCR transgenic mice and genetic reporter and fate-mapping models, the authors offer valuable insights into T cell reactivity and myeloid cell identity. The manuscript is well-written, with a particularly clear and engaging introduction. However, the sequencing analysis is relatively weak and could benefit from employing the Seurat pipeline for more robust interpretation. Additionally, the flow cytometry characterization of 'inflammation-associated microglia' (IAMs) appears to be affected by incorrect gating, and clarification is needed to distinguish BAMs from microglia. This could be addressed by leveraging the authors' Sall1 fate-mapping strain and incorporating immunofluorescence to differentiate CNS border and parenchymal populations. The discussion could be shortened. A comparison with work from the Rua or Movahedi labs on brain macrophages in infection would add important context. While there are areas for improvement, this study represents a valuable contribution to our understanding of immune responses during fungal CNS infections.

Response: We thank the reviewer for their careful critique of the paper and have addressed the comments in more detail below. Briefly, we used Partek software for analysis which uses the Seurat pipeline and have updated the methods to make this clearer. We have added several gating strategies as requested by the reviewer to provide further confidence in the separation between microglia and border macrophages, including an extensive analysis and new flow cytometry data that specifically analyses border macrophage populations (see responses below). We have shortened the discussion and focused this section on other infection studies, as suggested by the reviewer.

Specific:

Figure 1:

- 1f: A longitudinal analysis with quantification would help clarify the infiltration patterns of fungus-specific T cells. It seems unlikely that T cells initially infiltrate via brain capillaries across the BBB unless significant BBB damage occurs early on. Do the authors observe (fungus specific) T cells in the entry sites (leptomeninges and choroid plexus) between day 6-8? Could the authors do in vitro fluorescent labeling of TCRtg cells (alternatively, use the congenic marker as in Fig 2) and transfer them to infected mice to track them? A quantification of the imaging part(s) would be appreciated.

Response: We have included new data that analyses fungal-specific CnT.II cells in the meninges in revised Fig 2. We found small numbers of CnT.II present in this tissue as hypothesised by the reviewer. The total number of CnT.II cells in the brain is very low, even at late stages of infection. Therefore, these cells must be analysed using a high-throughput method such as flow cytometry, as detecting these rare cells by microscopy (even with fluorescent labelling) is not feasible. The small numbers present at early time points post-infection also hampered efforts to perform reliable longitudinal analyses, particularly in the meninges. We have now included a quantification of the CD4 imaging in the brain in the revised text (see lines 119-120).

Figure 2:

- c-d: Could the authors provide raw plots to show how they gated on CD44+ and proliferating T cells? Also it would be nice to look at those cells in the cLNs and get more information on the kinetics. As there is only one mouse analyzed on day 4 in the lung in c and d (but 2 in b?) it is not possible to say if the kinetics are more similar to the spleen or more similar to the lung. Typically, priming of fungus-specific T cells occurs in secondary lymphoid organs, with subsequent migration to infected tissues.

Response: Raw plots of our T-cell gating is shown in the new gating strategy figure, Fig S2. We added new cervical lymph node data to revised Fig 2. Day 4 data have been repeated and is now shown in Fig S3. We decided to move day 4 data to a supplemental figure and have revised the manuscript text so that this section focuses on the correlation between fungal burden and CnT.II infiltration, rather than time point post-infection. We took this decision since at day 4 post-infection there are too few CnT.II cells in the brain to accurately determine proliferation or activation, although we were able to detect small fractions of proliferating CnT.II cells with high CD44 expression in other tissues (see Fig S3).

Figure 4:

- The experiments with Tocky mice are intriguing. Could the authors show comparisons of brain, lung, and spleen using polyclonal TCR Tocky mice (Fig. 4e) to understand whether differences persist across tissues?

Response: We have added new data that shows the comparison of Tocky-positive CD4 T-cells between the brain, lung and spleen as well as example flow plots for each of these tissues to revised Fig 4.

Figure 5:

- Could the authors provide their gating strategy for the cell populations in Fig. S3A?

Response: We have added a new gating strategy figure, see Fig S11.

The analysis in Fig. S4 and 5 appears somewhat redundant to the flow analysis. Why were the time points split? A more integrated dataset could highlight temporal changes in myeloid cell composition more effectively. Heatmaps/dot plots of major clusters and volcano plots for DEG analysis between time points could simplify interpretation.

Response: The supplemental data the reviewer refers to (now Fig S6 and S7) show single-cell sequencing data, not flow cytometry. As explained in the main text, the time points were split between two experiments due to original experiment design and feasibility with number of animals needed for later time points. We opted to include basic analysis of each sequencing experiment as well as show the results of pooling the two datasets to demonstrate that there was no batch effect after correction. In-depth analysis of differentially-expressed genes was performed on the pooled dataset. We feel it would be redundant to perform the same depth of analysis for gene differences between clusters in individual datasets beyond what is already included in the paper. We have also not compared gene expression between time points as the study used single-cell RNAseq which has low sequencing depth. The advantage of this method is to examine variability between cell subsets, hence we have focused on analysing differences between cell clusters as is standard for most single-cell RNAseq studies.

Fig. S6 is not clearly explained in the text. It seems to depict BAM populations over time, but this should be introduced and interpreted in the results. In light of my concerns about the FC part of IAM characterization, a clearer BAM analysis could align better with the manuscript's main message.

Response: We have added a new paragraph to the main text to describe the data in Fig S8 (previously Fig S6) and have added new data showing flow cytometry characterisation of macrophage populations in uninfected and infected brains that align with our sequencing data.

The choice to focus on cluster 10 is unclear. Cluster 10 does not appear unique in the dot plot (e.g., lacking MHCII expression). Could the authors take an unbiased approach, identifying the microglia cluster most affected (e.g., clusters with the highest DEG activity on days 3 or 6)? This analysis could refine the study's focus.

Response: We chose to focus on cluster 10 since it was predominantly made up of cells that derived from infected brains, whereas other microglia clusters were more evenly distributed between uninfected and infected brains. This is a fair reason to focus on cluster 10, given that one of the study's aims is to examine changes in myeloid cells during infection. We feel that an analysis of number of DEGs between time points for the different microglia clusters is likely to yield erroneous results. This is because we think many of the microglia clusters are not biologically relevant and are only visible in the sequencing data due to artifacts introduced by expression of immediate-early response genes in some clusters (also see response below).

Comparing the DEGs with available signatures of isolation-induced transcript is commendable, but it is not relevant to their question as these signatures should be induced at all time points and, therefore, not contribute to DEGs?

Response: While the reviewer is correct that immediate-early response genes (IEGs) would be induced at all time points (as all samples were processed using the same methods), single-cell RNAseq has low depth (compared to bulk RNAseq) and therefore not all genes are picked up in every cell. These low-depth sequencing effects may cause artifacts in clustering data, and this is why we have been cautious in our interpretation regarding the number of microglia sub-clusters in the dataset. We have clarified this point in the text and added a reference which further explains this phenomenon.

Figure S8:

Line 246: Could the authors clarify how the genes in cluster 10 were identified? This information is missing.

Response: This information has been added.

Line 248: The authors state they aim to examine how CD4 T cells influence cluster 10. However, as cluster 10 only appears on day 6, this question is challenging to address. Comparing microglia at day 3 (pre-T cell influx) with day 6 could be more informative. The introduction to this analysis might be better placed in a later section (e.g., Figure 8).

Response: We have removed the reference to the T-cells in the introduction to the sequencing analysis to improve the flow of the manuscript. We have not focused on time point for analysis, instead focusing on heterogeneity of cell subsets, as outlined in our earlier responses.

Panels S8A–B: The analysis here is somewhat unconventional. Could the authors compare the top 10–20 DEGs between DAMs and IAMs? More clarity is needed on what makes cluster 10 distinct

Response: Our analysis encompasses a comparison of both upregulated and downregulated DEGs, across all detected genes in the different datasets. This is more comprehensive than comparing only 10-20 genes.

Figure 6:

- Line 281: The claim that IAMs express Mrc1 (CD206) is not apparent from the cluster description in Fig. S3.

Response: We do not claim Mrc1 is highest in the IAMs cluster when comparing across all clusters in the dataset (which would include Mrc1-high border macrophages). Rather, amongst the microglia clusters specifically, Mrc1 is highest in the IAMs. Below is a table of the mean values of Mrc1 in microglia and non-microglia clusters for the reviewer's reference. We have also clarified this point in the main text (lines 324-328).

	Mrc1 mean expression
Microglia clusters	
1	0.081172
2	0.117266
3	0.099854
4	0.102270
5	0.066485
6	0.076947
9	0.161935
10 (IAMs)	1.752870
12	0.033021
Non-microglia clusters	
7 (macrophages; border and inflammatory)	5.70202
8 (neutrophils)	0.078695
11 (monocytes)	0.864328
14 (macrophages; border)	5.25074

The flow cytometry gating in Fig. 6B likely captures BAMs, not microglia (BAMs typically express CD204 and CD206 at high levels, while microglia do not). To confirm microglia identity, the authors should perform immunofluorescence (IF) to localize IAMs in the parenchyma and demonstrate Sall1 expression in their IAM gate. Raw flow plots supporting Fig. 6f,g would be helpful.

Response: In new Fig S11, we provide several plots that demonstrate that IAMs and CD206+ macrophages are distinct populations in our flow cytometry experiments. This was confirmed using Sall1-reporter mice, which we used to validate our method of separating microglia and macrophages (as shown in Fig 6). We have also performed staining for proliferating cells using the Ki67 marker in brain sections and found these cells localised in the parenchyma near sites of infection and were increased in the infected brain compared to uninfected, aligning with our flow cytometry data. This new IF data is shown in new Fig 9.

Given the rarity of cluster 10 (<10%), preselecting for microglia using Sall1 expression might be necessary before identifying CD204+ subsets. Could the authors clarify whether cluster 10 is defined by CD206 expression in their scRNA-seq data?

Response: Our gating strategy (see new Fig S11) includes a pre-selection step to separate microglia and macrophages based on differential CD45 and CX3CR1 expression, a commonly used and well-accepted tactic in the neuroimmunology field. CD206 and Msr1 double-positive IAMs are therefore identified within the microglia gate, after excluding CD45^{hi} cells which includes CD206^{hi} macrophages. Panels c-e in Fig S11 show examples of back-gating and comparisons between these populations to demonstrate that they are distinct populations in our flow cytometry experiments.

As shown in the table we provide in response above, CD206 expression is higher in IAMs relative to other microglia (and was identified as a significantly differentially expressed gene, see Table S1; fold change 40.8, $P=4.85 \times 10^{-82}$), but is lower compared to macrophages.

Figure 7:

The gating strategy here should be included in the supplement. It appears the data may reflect BAMs rather than microglia. Repeating the analysis using Sall1 fate-mapping mice and a more precise gating strategy would help.

Response: See earlier responses; we have added the gating strategy in Fig S11 with additional plots to demonstrate that microglia and macrophages are distinct populations.

Please include the gating on GFP+ IAMs and non IAMs

Response: Gating for these populations is shown in Fig 7a. We have also added plots showing ungated brain samples, comparing with non-fluorescent fungal-infected controls, in Fig S11f.

The "microglia-depleted brain" image is confusing, as this analysis is not described in the text. Is this a leftover of their previous publication? <https://www.nature.com/articles/s41467-023-43061-0#MOESM1>. If relevant, it should be clarified.

Response: This image and its purpose are described in the figure legend. We have added the title 'staining control' to the figure to make this clearer.

Using the Sall1 fate-mapping strain to visualize fungal uptake would enhance this figure. Including comparisons at brain borders (e.g., CD204+/CD206+ BAMs) versus parenchyma would be valuable. Quantification of these images is also recommended.

Response: We no longer breed the Sall1 fate-mapping strain so are unable to perform the IF experiments requested. Instead, we have provided the gating strategy (Fig S11) and IF of Ki67+ cells (see earlier responses) to further convince that IAMs and macrophages are distinct populations and that proliferating cells are localised near sites of infection in the parenchyma.

The references to panels 7f and 7g seem incorrect; they likely refer to panels 7g and 7h, respectively. Could the authors clarify?

Response: This has been corrected.

The absence of Arginase induction in microglia from cpl1 mutants is unsurprising. A comparison between microglia and non-microglia populations could support their conclusions.

Response: We created a new subsection in the results and added new data showing the frequency of arginase production in MHCII^{hi} and MHCII^{lo} brain macrophages. These experiments are shown in new Fig 8.

Figure 8 :

Please include gating strategies for IAMs in Fig. S9.

Response: See earlier responses; a full gating strategy has been provided in Fig S11.

Panels 8b–g seem to analyze BAMs rather than microglia (low CX3CR1 and high CD206 expression). This should either be repeated as described above or focused on BAMs in sequencing and imaging analyses.

Response: The plots shown in Fig 9 (previously Fig 8) are pre-gated on CD45^{int}CX3CR1^{hi} cells (i.e. microglia, the CX3CR1⁺CD45^{hi} cells which include BAMs and other macrophages have been gated out). Defining microglia in this way aligns with our Sall1-reporter analysis (see responses above and new Fig S11) and extensive published data.

Line 368: The statement on IAM-driven inflammation and its link to CD4 T-cell infiltration is not addressed experimentally

Response: This line refers to published work in the context of autoimmunity. Our data demonstrates that IAMs are a proliferative subset, and that microglia proliferation is dependent on CD4 T-cells, which we have shown in multiple models.

The role of CD4 T cells is only while the role of IFN γ is not addressed in the manuscript. Consider rephrasing this text to align with the study's flow.

Response: We removed the reference to IFN γ as suggested by the reviewer.

Could the authors evaluate what happens when microgliosis is blocked, and whether this benefits *C. neoformans* infection and/or the clinical outcome?

Response: We considered several approaches to block microgliosis but many of these came with caveats. The drug ganciclovir, mentioned in our discussion, has been shown to block microgliosis but had also been shown to affect CD4 T-cell responses, meaning that any experiments of this kind would be difficult to interpret. Indeed, we show in two independent models that a lack of lymphocytes results in reduced microgliosis and the impact of this on clinical outcome.

Is the fungal clearance in the brain (and periphery) different in RAGKO mice?

Response: We see no difference in fungal brain burdens between wild-type and RAG KO mice, infected with two different strains of *C. neoformans*. This new data aligns with our data using CD4 depleting antibodies and is included in new Fig 10. We have opted only to show brain burdens here since we do not display burden data for other organs in other experiments as the main focus of the paper is the brain.

The title claims that brain infiltrating CD4⁺ T cells drive microgliosis. Can this also be supported by IF? Do the authors see differences in the depletion of intravascular and brain-invading CD4 cells (using intravascular labeling by CD45 or CFSE as the authors used before)?

Response: We're unable to specifically deplete brain-infiltrating and intravascular populations of CD4 T-cells since it's possible that brain-infiltrating populations may originate from intravascular populations, particularly at later stages of infection when barrier damage may occur. We're therefore not able to perform the experiment the reviewer suggests. Instead, we provide new images and data of Ki67⁺ cells in brain sections from wild-type and Rag KO animals, which showed a reduction in Ki67 labelling in the lymphocyte-deficient mice, reflecting our flow cytometry findings.

Minor:

- L50 abbreviation for iNOS missing

Response: This has been corrected.

- L213, the authors write they used n=2-5 mice per time point, but the legend says 3-4. What is it? The exact numbers would be more informative as it is only 3 time points.

Response: The figure legends have been updated to make n numbers clearer. The 2-5 mice referred to the single-cell RNAseq experiment, whereas the 3-4 mice referred to a flow cytometry experiment.

- L261 type: "RMA" instead of RNA

Response: This has been corrected.

- L336: NO has been abbreviated before

Response: This has been corrected.

- In the methods, the authors write about Sall1CreER Csf1r1fl mice, but this strain has not been used

Response: These references have been removed.

Reviewer #2 (Remarks to the Author)

This is a mouse study describing in great detail the kinetics of the CNS infection with *Cryptococcus neoformans* H99 in a mouse model. This study represents the collaborative work of 3 important fungal immunology groups and used the newest tools to characterize the immune response in this model at 3 distinct time points.

The strengths of this paper are the quality of data and writing and the masterful use of relatively novel tools. The weakness is that this paper represents a purely descriptive work without testing any specific hypothesis. While the authors discuss some possible future directions, such as testing the role of TLR4, no comparisons have been made beyond the detailed model characterization.

There is also a major concern that a single strain of *C. neo* is used (H99), which this group already showed is capable of extensive growth in microglia in contrast to other strains of *C. neo*. *Nat Commun.* 2023 Nov 8;14(1):720

Thus, some of the conclusions about the weak fungicidal capacity of activated microglia could be biased by the choice of this strain.

Response: We have added new data which utilises an additional strain of *C. neoformans*, Zc1, which is in the same molecular type as the H99 reference strain but distantly related. We found a similar number of proliferating microglia in the brain of Zc1-infected mice, and that the generation of this population is similarly dependent on lymphocytes as we observed significant reductions in Rag-deficient mice infected with Zc1. This new data is included in Fig 10. Assessing fungicidal capacity in vivo across different strains would require making fluorescent reporter strains in the different backgrounds, which we feel is out of scope of the current work. Instead, we have added a line to the discussion stating the caveat the reviewer raises.

Other comments:

The adoptive transfer of TR Tg cells was used to demonstrate that brain recruitment of the antigen-specific T-cells is delayed compared to non-specific T-cells. However, the antigen-specific T-cells are transferred in a low proportion compared to the host T-cells on day -1, and we don't know how long these cells require to populate the lymphatic system and adapt before they assume their physiological functions in vivo. So there is a possibility that the delay is a cell-transfer artifact. Would we obtain a similar outcome if the non-specific cells were transferred to the TCR transgenic mouse?

Response: We agree that adoptive transfer models have important caveats as outlined by the reviewer. We therefore felt that the experiment suggested would be subject to similar issues. Instead, we used a previously published MHCII-tetramer loaded with Cda2 peptide to demonstrate infiltration of fungal-specific CD4 T-cells into the brain of *C. neoformans* infected mice that have normal T-cell repertoires. This new data is shown in revised Fig 1. We've also softened conclusions regarding the delay in responses and moved the day 4 time point data to the supplemental figures, instead focusing on the correlative relationship between fungal burdens and T-cell responses (see also response to reviewer 1).

Some human specimen data validating at least some of the findings presented here would greatly strengthen the paper.

Response: We agree with the reviewer that human specimen data would be interesting and help strengthen our findings. However, human brain samples from autopsy of cryptococcal meningitis patients are scarce (particularly in the UK where we are based) and we haven't been able to source these tissues in a timely manner for this publication (although we are working towards this for a future manuscript). We have added a line to our discussion to highlight this limitation of the study.

The discussion covers a lot of different topics/areas not tested in this paper.

Response: The discussion section has been reduced (see also response to reviewer 1).

Reviewer #3 (Remarks to the Author)

In this manuscript by Hain et al, the authors describe the role of inflammatory microglia in a mouse model of cryptococcal meningitis. Although the data are interesting, there are some areas the authors need to address.

Major issues:

1. As the authors mention multiple times in the manuscript, the main population of humans affected with cryptococcal meningitis are AIDS patients, who have an extremely low CD4+ T cell count. Although the data showing CD4+ T cell induction of IAM proliferation (Figure 8) are very interesting, this is somewhat counterintuitive to the situation in AIDS patients, where most symptomatic cryptococcal meningitis is found. Although IRIS is an issue, many clinical studies point to an excessive pro-inflammatory response due to reconstitution. It is difficult, with this study or others, to determine whether or not the reconstitution of CD4+ T cells and their interaction with microglia are specifically involved in IRIS.

Response: In IRIS, the excessive pro-inflammatory response that drives pathology is thought to be driven by CD4 T-cells. In our study, we show that CD4 T-cells have a complex relationship with microglia and are responsible for driving inflammatory proliferation of these cells during infection in mice. We feel this raises important questions for future studies examining inflammatory signatures in patients with IRIS and may be important when considering how therapies that boost CD4 T-cell function affect microglia phenotype in the context of cryptococcal meningitis. We have highlighted the caveats of our model to our discussion and made clear the need for further clinical studies that examine the interactions between cell types (see also response to reviewer 2).

2. The intravenous model of infection, while used widely, is not truly representative of trafficking of *C. neoformans* from the lung to the brain to induce meningitis. Since the authors did not compare the brain microglia outcomes from a pulmonary infection to the i.v. infection, it is difficult to assess the actual implications of cell involvement during the natural course of infection, especially when myeloid cells from the lung (which may traffic during a pulmonary infection) are not involved in the i.v. model.

Response: We performed new experiments using the intranasal route of infection and examined fungal burdens, microglia proliferation and numbers of IAMs in the brains of mice infected via this route. This new data is shown in Fig 9. In this model, mice die from overwhelming lung infection before meningitis or significant brain burdens develop. This limits the levels of brain infection and inflammation that we can observe with this infection route. Therefore, while we were able to detect microglia proliferation which largely mapped to IAMs, this was milder in intranasally infected mice compared to the intravenously infected mice, likely aligning with the lower fungal burdens in the brains of these animals.

3. While most experiments seem to have replicates, some experiments were only conducted once. Some had different numbers of experiments depending on the time point (for example, Figure 2 shows 1 experiment for D4 vs 2 experiments for D0 and D7), which makes it difficult to assess reproducibility of the results as well as consistency of the experiments (for example, if the D7 and D4 time point were from different adoptive transfers and different fungal inocula). Therefore, it is difficult to assess some of the results (especially those in Figure 2) due to the inconsistency and lack of reproducibility of some of the data shown.

Response: See also response to reviewer 1. Day 4 data have been repeated and shown in new Fig S4.

4. In the studies examining IAM fungicidal activity, it is difficult to discern the antifungal activity based on the experimental setup. If the IAM and non-IAM cells contain different numbers of *C. neoformans* cells prior to removal and plating of the cells ex vivo, then it is not possible to calculate killing capacity of those cells. For example, in Figure 7a, the % infected cells is shown, but that does not mean that each cell is only infected with a single cryptococcal cell. Even though the cells are sorted for GFP+ cryptococcal cells prior to plating, that does not mean that those cells contain the same number of GFP+ cells. Therefore, these data are impossible to assay for antifungal activity using these methodologies.

Response: We have performed new experiments in which we used an imaging cytometer to image infected IAMs and non-IAMs, and then quantified the number of yeast per cell. This analysis showed that the majority of microglia have one yeast per cell (68% of IAMs and 89% of non-IAMs). There was a

significant increase in the IAMs population of microglia hosting multiple yeast, which was observed in ~30% of the population. This new data is shown in revised Fig 7. Despite this increase, we feel our conclusion of reduced fungicidal capacity of IAMs is fair given this difference in yeast per cell could not fully account for the large difference observed in our fungal viability assay.

Minor issues:

1. In the legend for Figure 5, the number of animals per group per time point and the number of experiments is missing.

Response: This information has been added.

REVIEWERS' COMMENTS

Reviewer #1 (Remarks to the Author):

The manuscript has improved following revision, and most of the points raised during review have been addressed. Thank you for your thoughtful responses and the additional analyses.

Reviewer #2 (Remarks to the Author):

The manuscript has been revised and supplemented with additional data, improving greatly over the previous version. There is one major comment regarding the results linked to Figure 8 and a few minor issues (recommended edits).

Major:

LN386-407; The conclusion that CPL-1, as correctly labeled in methods (elsewhere spelled as CLP-1) promotes arginase expression in IAM and intracellular residence is not fully supported.

* LN 388: While poor fungal killing was linked to Arg1, causality has never been established here or in other studies. This would require Arg1 inhibitor or Arg1 knockdown to be used, and showed that the cells kill the fungus better when Arg1 is inactive.

* LN 397-402 The scenario presented here is only one of the options. The data shows that the mutant is 2 orders of magnitude lower in the brain; therefore, even if the mutant triggers Arg1, we are not likely to see it due to the sparsity of the fungus (1% of what we see in the WT).

* Quite contrary, the data on Fig 8D shows that despite the very low antigen load, up to 30% of MHCII-hi cells can be Arg1+, and on average their Arg1+ frequency in MHC-hi is similar to that in MHCII-lo cells with the WT, which are only 5% infected.

* We also do not know why there are fewer mutants in the brain: did it not cross the BBB, did it die due to reduced adaptation to the brain environment (such as very low glucose or copper), or other fitness problems?

Therefore, from this data, we cannot conclude that cpl-1 was responsible for Arg1 induction, and this, in turn, was the cause of its lower CFU.

This part is not essential for the flow of the manuscript, while the data is insufficient to sustain the author's conclusion without much additional work to clarify all the issues above. Thus, most likely, we need this part to be removed unless additional convincing data is included.

Response: We have removed the CPL-1 data from the manuscript as suggested by the reviewer to improve the flow and as discussed with the editorial team.

Minor

LN 50-54 The cited study “2” is a review and original articles should be cited; also, while the information about IFN-g is correct, the role of NO in fungal killing is contextual. Multiple studies showed it to be dispensable or less important than other fungicidal macrophage mechanisms. eg: PMID: 11092381, 11907100, 38841113. Thus, this statement about NO playing a major role should be softened or removed.

Response: We have softened the text and included a citation to the primary research article as suggested by the reviewer.

LN 147 “proliferated” fits better than “proliferating” here, since we don’t know at which site proliferation occurred based solely on low CFSE.

Response: We have updated the text as suggested by the reviewer.

Reviewer #3 (Remarks to the Author):

The updated manuscript has addressed all review criteria.